# Impacts of environmental and socio-economic factors on emergence and epidemic potential of Ebola in Africa

David W. Redding [1]*, Peter M. Atkinson[2], Andrew A. Cunningham [3], Gianni Lo Iacono [4], Lina M. Moses [5], James L.N. Wood [6] & Kate E. Jones [1,3]*

Recent outbreaks of animal-borne emerging infectious diseases have likely been precipitated by a complex interplay of changing ecological, epidemiological and socio-economic factors. Here, we develop modelling methods that capture elements of each of these factors, to predict the risk of Ebola virus disease (EVD) across time and space. Our modelling results match previously-observed outbreak patterns with high accuracy, and suggest further outbreaks could occur across most of West and Central Africa. Trends in the underlying drivers of EVD risk suggest a 1.75 to 3.2-fold increase in the endemic rate of animal-human viral spillovers in Africa by 2070, given current modes of healthcare intervention. Future global change scenarios with higher human population growth and lower rates of socio-economic development yield a 1.63-fold higher likelihood of epidemics occurring as a result of spill-over events. Our modelling framework can be used to target interventions designed to reduce epidemic risk for many zoonotic diseases.

---

[1] Centre for Biodiversity and Environment Research, Department of Genetics, Evolution and Environment, University College London, Gower Street, London WC1E 6BT, UK. [2] Lancaster Environment Centre, Lancaster University, Bailrigg Lancaster LA4 1YW, UK. [3] Institute of Zoology, Zoological Society of London, Regent's Park, London NW1 4RY, UK. [4] School of Veterinary Medicine, University of Surrey, Guildford, UK. [5] Department of Global Community Health and Behavioral Sciences, Tulane University, New Orleans, LA, USA. [6] Department of Veterinary Medicine, Disease Dynamics Unit, University of Cambridge, Cambridge, UK. *email: d.redding@ucl.ac.uk; kate.e.jones@ucl.ac.uk

L ittle is known about how the majority of human infectious diseases will be affected by predicted future global environmental changes (such as climate, land use, human societal and demographic change)[1–5]. Importantly, two thirds of human infectious diseases are animal-borne (zoonotic)[6] and these diseases form a major, global health and economic burden, disproportionately impacting poor communities[7,8]. Many zoonotic diseases are poorly understood, and global health responses to them are chronically underfunded[9]. Our knowledge gaps and the need for improved forecasting of zoonotic disease risk were starkly illustrated by the 2013–2016 Ebola outbreak, which was unprecedented in terms of size, financial cost, and geographical location[10,11].

Ebola virus disease (EVD) was first identified in 1976, and since then there have been ~23 recognized outbreaks[12], predominantly within central Africa. EVD is caused by any one of four pathogenic strains of Ebola virus: Zaire (EBOV), Sudan (SUDV), Taï Forest (TAFV), and Bundibugyo (BDBV). It presents as a non-specific febrile illness that can cause haemorrhagic fever, often with a high case fatality rate in diagnosed patients[13]. Some Old World fruit bat species (Family Pteropodidae) have been suggested as reservoir hosts[14], however, while there is limited direct evidence, they are strong candidates to play a key role either as a reservoir or amplifying host[15,16]. In areas with EVD, there are frequent direct and indirect human-bat interactions, e.g., via bush meat hunting and during fruit harvesting[17], presenting numerous opportunities for bat-to-human pathogen spill-overs to occur. Additionally, a third of known zoonotic spill-overs have been connected to contact with great apes and duikers, although there is no evidence that these species act as reservoir hosts[10]. It is clear, however, that once spill-over occurs human social factors such as movement and healthcare responses greatly influence the cumulative outcome of an outbreak[18]. For instance, previous work has highlighted the importance of family interactions[19], funeral practices[20] and differential transmission rates in hospitalized individuals[18].

Many attempts to understand Ebola outbreak dynamics have focused on mechanistic modelling approaches of human-to-human transmission post spill-over from animal hosts[13,18,19,21–24]. Mechanistic, or process-based, models are ideal for capturing epidemiological characteristics of diseases and, importantly, testing how disease outbreaks might be impacted by intervention efforts[10]. One downside is that mechanistic models of zoonoses often do not incorporate spatially heterogeneous ecological and environmental information, such as the environmental differences leading to variation in host suitability[25]. In this context, correlative, or pattern-based, models (e.g., MaxEnt, Boosted-Regression Trees) have been used to simultaneously capture the spatial risk of both zoonotic spill-over and subsequent human-to-human infection[12]. For some spatially-explicit analyses of Ebola, there have been attempts to incorporate spatial patterns of human populations and/or air or other transportation networks[26–28], but few studies have considered whole-system analyses for major epidemic zoonoses across the whole endemic disease area. Like other rare or poorly-sampled diseases, Ebola suffers from limited data availability, meaning approaches using pattern-finding, correlative analytical techniques to find robust models are at a disadvantage[29].

In 2014, a spill-over in Gueckedou district, Guinea of Ebola-Zaire virus led to an EVD outbreak ~100 times larger than any of the previous 21 known outbreaks[30]. Such epidemics have a disproportionate impact on the affected societies. For example, the World Bank estimates a cost of US$2.2 billion to the three most affected countries[31] due to, amongst other drivers, widespread infrastructure breakdown, mass migration, crop abandonment and a rise in endemic diseases due to overrun healthcare systems. Recent work has uncovered the human-to-human transmission

patterns underlying this outbreak, using case[32] and genomic data[30] to demonstrate that EVD spread can be successfully predicted by a dispersal model that is weighted by both geographic distance and human population density. Attempting to understand zoonotic epidemic risk, however, using a human-only transmission model and without incorporating host ecology would inevitably lead to areas with high human density and connectivity being identified as the regions with the highest risk, despite some areas of these lacking competent hosts. Therefore, to model both the spatial variation in spill-over risk and, concurrently, the likely progression of subsequent outbreaks in human populations, we need to take a system-dynamics modelling approach[1,33]. Key non-linear feedbacks can also be captured, for example, the trade-off between increasing human populations and any potential loss of reservoir host species through anthropogenic land-use conversion. We can use this approach to create a methodological test-bed to design adaptive vaccination and epidemic preparedness regimes across Africa that are future-proof with respect to changing ecological and human landscapes[34].

Here, using a disease system-dynamics framework (Fig. 1), we develop a discrete-time, stochastic epidemiological compartmental model (Environmental-Mechanistic Model or EMM, Fig. 2) incorporating spatial environmental variability to simulate present-day spill-over and subsequent human-to-human transmission using the example of Zaire Ebola virus (EBOV) (the strain responsible for the 2013–2016 outbreak in West Africa). We run the model ~20,000 times with randomly-sampled starting conditions and critically examine the emergent patterns of infections to discover which areas of the world are most at risk, and explore currently unsampled effects of Ebola epidemiology in Africa, and potential spread of EBOV at global scale based on flight data. As our model is flexible in terms of inputs, we can then examine the possible trend of Ebola epidemiological patterns by projecting to a future time-point, 2070[35–37] under a variety of integrated global change scenarios[38]. We compare the model output from three Representative Concentration Pathways (RCP) scenarios of increasing greenhouse gas concentrations: GCAM-RCP4.5 ('High Climate Mitigation'), AIM-RCP6.0 ('Low Climate Mitigation'), and MESSAGE-RCP8.5 ('Business as Usual Emissions')[39], and three possible socio-economic development scenarios (Shared Socio-economic Pathways or SSP), ordered by increasing human population density and reduced regional socio-economic cooperation: SSP1, SSP2 and SSP3 (termed respectively 'Sustainable', 'Middle of the Road' and 'Regional Rivalry' Development). Using these scenarios as reasonable boundaries of possible future global change, we then identify the prevailing direction of changes to the spatial patterns of risk of EBOV outbreaks and epidemics across Africa, and subsequent importations of disease cases across the world via airlines.

## Results

**Present day patterns.** Our EMM simulation for present day EBOV-EVD risk correctly identified areas of observed outbreaks as high risk, such as Democratic Republic of Congo, Gabon and the 2013–2016 outbreak in West Africa, but also identified some areas where EVD has not been reported, including most countries in West and Central Africa, especially Nigeria and Ghana (Fig. 3a). Our model outputs predict the observed pattern of outbreaks with high accuracy (AUC = 0.83 ± 0.012–95% confidence interval (CI) from 10,000 repetitions of 1000 random pseudo-absences), which is significantly higher than 10,000 randomised versions of the risk surface (AUC = 0.53 ± 0.028–95% CI), a pattern which held irrespective of how the endemic area was defined (Supplementary Fig. 1). Since the modelling exercise was run in 2018, our model has successfully predicted subsequent

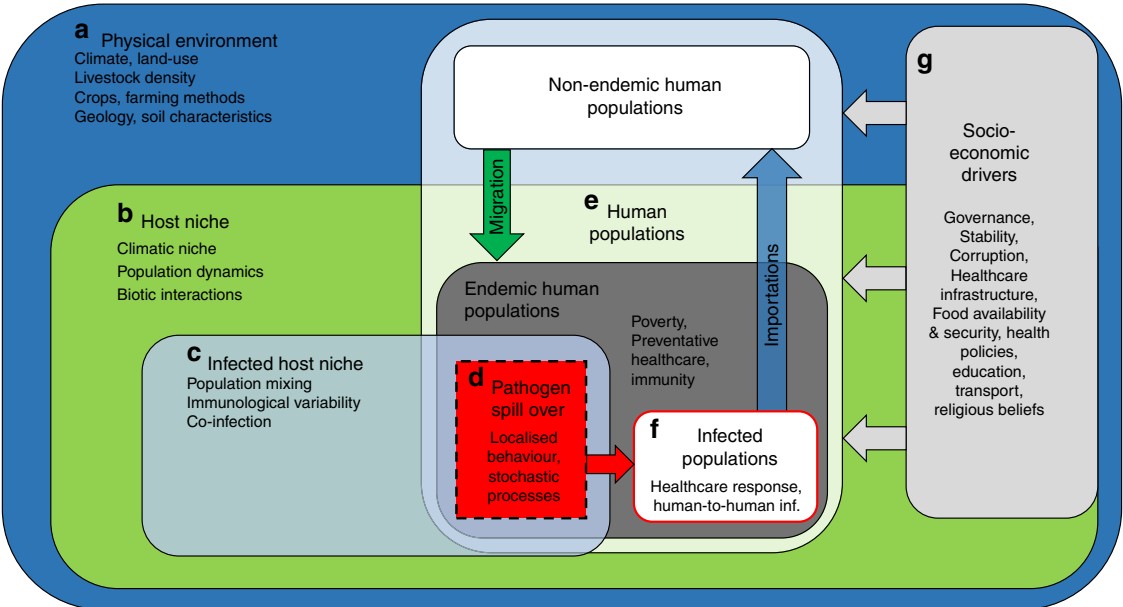

**Fig. 1** System-dynamics model of zoonotic disease transmission. Letters **a–f** indicate major system components, arrows showing links, and key sub-components in smaller font. Within the global physical environment (**a**), both the host niche (**b**) and infected host niche (**c**) are nested subsets, which all vary over a relatively slow time-scale. Endemic human populations are nested within the global human population (**e**), with human socio-economic factors (**g**) affecting all human populations. Spill-overs happen in the fast-moving spatial and temporal interface of these two nested systems (**d**), where both infected hosts, susceptible people and spill-over specific factors occur, resulting in infected human populations (**f**)

outbreaks (Equateur & Kivu provinces, Democratic Republic of Congo) and indicate they occurred in areas at high risk of larger size (1500+ cases) outbreaks (Fig. 3c). Our predictions also replicate the pattern where outbreaks in the Gabon hotspot are likely to be smaller in terms of total case size, with those in the east and west having the potential for larger outbreak sizes (Fig. 3). Our modelling approach, which focuses on establishing the fundamental conditions where EBOV could reasonably spill-over, without using case data, provides further support that the at-risk area for EBOV-EVD spill-overs to occur (Fig. 3b; Guinea-Bissau, Liberia, Sierra Leone, Ghana, Togo, Benin, Nigeria, Cameroon, Gabon, Equatorial Guinea, Congo, Central African Republic, Ruanda, Burundi, Kenya and Tanzania) is much larger than the area made up of just those countries that have reported disease outbreaks thus far. We note that our risk map also identified high-risk areas that are endemic for the other EVD strains, likely due to similar transmission pathways and reservoir host characteristics (Fig. 3a). When comparing the mean number of spill-overs across Africa per year, present day simulations were higher (2.464 spill-overs per year 95% CI 2.361–2.567) than the mean historical number over the last 40 years: 0.75 (95% CI 0.695–0.905), reflecting the increase in human population in these regions during this time period. High risk of Ebola case importation outside Africa, using the current network of airline flights, was seen in China, Russia, India, the United States as well as many high-income European countries (Fig. 4), a similar pattern to the importations seen during the 2014–2016 outbreak (United States, Spain, Italy, United Kingdom). (Fig. 4).

Similar to historic data (Supplementary Fig. 4), the distribution of the final size of the simulated outbreaks was multimodal with distinct peaks at very low numbers (less than 3 cases) and medium outbreaks (3–1500 cases) (Supplementary Fig. 4). Under extensive simulation, the most common outbreaks were very small, at odds with the observed data, with singleton or two-person outbreaks frequently observed. We are also able to explore the lower probability areas of the distribution effectively and, unique to the simulation data, there is a third peak of outbreaks

(here we term 'epidemics') with high, to very high, numbers of cases (1500+). This threshold of assigning an outbreak with greater than 1500 cases as an epidemic also corresponds to the top 1 percentile of a log-normal distribution approximating the variation in pre-2016 observed outbreak sizes (~1538 cases per year). Of the ~2500 simulation runs for present day conditions, epidemics (>1500) occurred approximately in 5.8% of the yearly simulations, with catastrophic epidemics (>2,000,000) occurring in around 2.3% of simulations, or once every 43.5 years given current conditions. From the sensitivity testing, the key parameters that affected outbreak size were illness length and $R_0$, which positively increased case numbers (Supplementary Fig. 5A), whereas the annual spill-over rate (Supplementary Fig. 5B) was most impacted by the spill-over rate constant (strongly positive), shape of the poverty/spill-over curve (weakly positive), and by host movement distance (weakly negative).

**Future trajectories**. Our future EMM simulations suggest that, in most scenarios, given current projected patterns of global change, there will be a general, ongoing increase in Ebola incidence over time. For example, we estimate an annual increase in the maximum area impacted by the disease from 3.45 million $km^2$ to 3.8 million $km^2$ under the worst-case scenario by 2070, with increases in the maximum area seen under all future scenarios. The maximum area where just spill-overs could occur, however, increased by just 1% under the GCAM-RCP4.5 SSP1 (High Climate Mitigation + Sustainable Development: 2.01 million $km^2$), when compared to present day (Fig. 3b: 1.99 million $km^2$), but increased by 14.7% from present day under the MESSAGE-RCP8.5 SSP3 (Business as Usual Emissions + Regional Rivalry Development: 2.29 million $km^2$) scenario. Conversely, the total area where epidemics could start decreased under the GCAM-RCP4.5 SSP1 by 47% (0.444 million $km^2$), when compared to present day (Fig. 3c: 0.836 million $km^2$), but again increases under AIM-RCP6.0 SSP2 (Low Climate Mitigation + Middle of the Road Development) this time by 20.5%, and then by 34% under

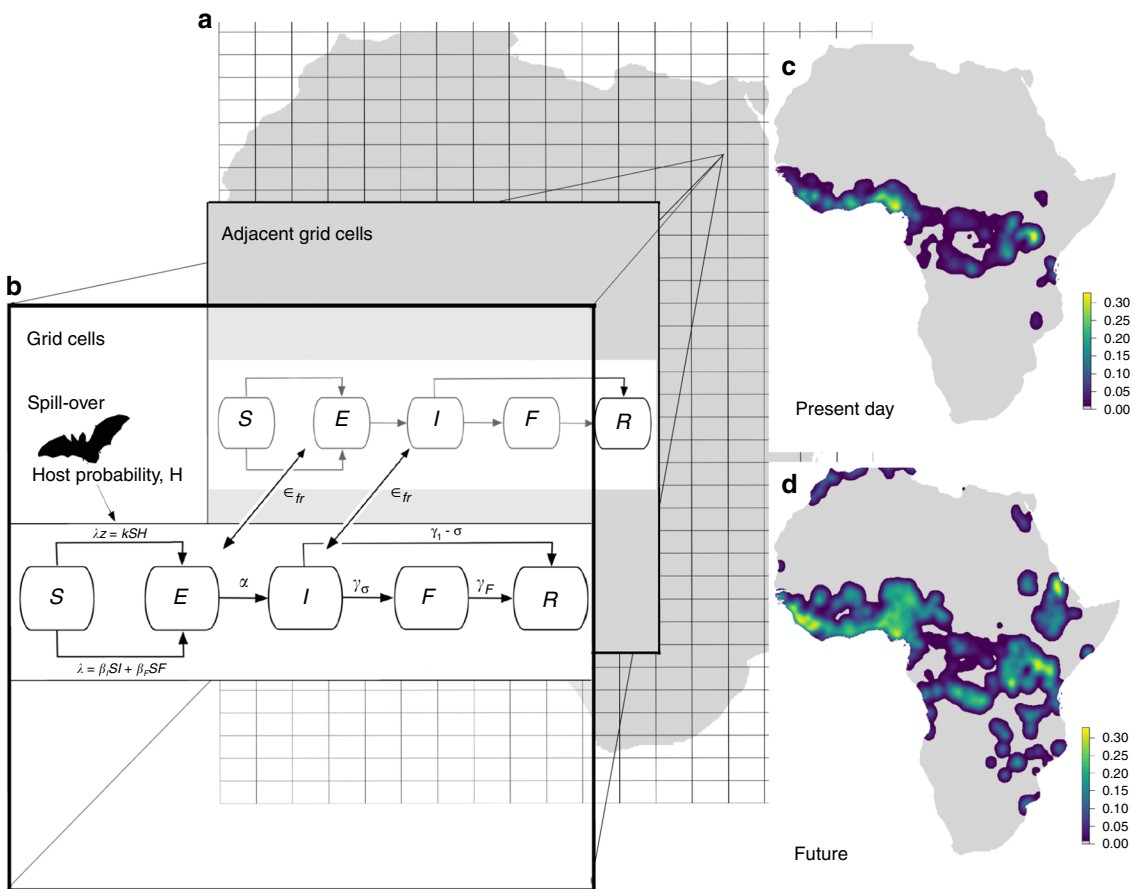

**Fig. 2** Environmental Mechanistic Model (EMM) EBOV Simulation Schematic. Within 0.0416° (5.6 km at equator) grid cells across the globe (**a**), we used a *SEIFR* (*Susceptible, Exposed, Infectious, Funeral* and *Removed*) disease compartmental model (**b**), to estimate the number of people in each compartment. *S–E* transmission rate was determined for each grid cell by calculating the force of zoonotic infection (between hosts and humans) $\lambda_z$, and within human populations $\lambda$ (see Methods). Travel of exposed or infectious individuals between grid cells occurred across existing road and flight transport networks, with transmission rate $\epsilon_{fr}$. Mean transition rates used as the starting parameters for simulations were as follows: $\alpha$ for *E–I* was calculated as the reciprocal of incubation time in days ($\alpha = 1/7$), $\gamma_\sigma$ (*I–F* transition rate) was the product of the probability of the reciprocal of days infectious ($\gamma = 1/9.6$) and maximum poverty-weighted case fatality rate ($\sigma = 0.78$), $\gamma_{1-\sigma}$ (*I–R* transition rate) was the product of the probability of the reciprocal of days infectious ($\gamma = 1/9.6$) and probability of recovering (1-$\sigma$), and $\gamma_F$ (*F–R* transition rate) was the reciprocal of the burial time of 2 days. Each simulation was run 2500 times for 365 days, only including grid cells containing a human population. The total number of people in each compartment per grid cell, per day from each simulation was then used to calculate the total number of index and secondary cases and mapped spatially (**c**). Bioclimatic, land use and demographic conditions were then changed to predicted values for 2070 to estimate changes to $\lambda$ and $\lambda_z$, and the simulations repeated to investigate impacts of global change on disease (**d**)

the MESSAGE-RCP8.5 SSP3 scenario. Country-level patterns showed a trend of a general decrease in EVD risk for the better-case scenarios (e.g. High Climate Mitigation + Sustainable Development) towards an increase in risk, under the worst scenarios (e.g. Business as Usual Emissions + Regional Rivalry Development), with this pattern especially pronounced in West Africa (Fig. 5).

The increases seen in the area affected is mirrored by greater total numbers of spill-overs experienced in future scenarios, with the greatest increase seen under the MESSAGE-RCP8.5 SSP3 scenario at 7.92 (CI 7.62–8.19) spill-overs per year. Spill-over numbers increased with greenhouse gas concentrations (represented here by the RCP value) with a mean 0.257 spill over a year increase between the GCAM-RCP4.5 SSP2 (High Climate Mitigation + Middle of the Road Development) and AIM-RCP6.0 SSP2 scenarios, and a mean 0.343 spill over a year increase between the AIM-RCP6.0 SSP3 (Low Climate Mitigation + Regional Rivalry Development) and MESSAGE-RCP8.5 SSP3 scenarios. Greater increases were seen, however, with SSP change, with a mean 1.297 spill over a year increase between

GCAM-RCP4.5 SSP1 and GCAM-RCP4.5 SSP2 scenarios and a mean 1.475 spill over a year increase between AIM-RCP6.0 SSP2 and AIM-RCP6.0-SSP3. In general, the probability of the index cases resulting in small outbreaks reduced in future environments, whereas the chance of epidemics increased (Fig. 6). For instance, the proportion of epidemics per year (>1500 cases) decreased in the GCAM-RCP4.5 SSP1 to 3.43% (from 5.8% in present day) but increased in all others, with AIM-RCP6.0 SSP3 gaining the greatest number, with epidemics in 9.5% of all simulations. The number of catastrophic epidemics (>2,000,000), generally increased with both RCP and SSP values up to 3.43% and 3.54% for the AIM-RCP6.0 and MESSAGE-RCP8.5 SSP3 scenarios respectively, but again saw a decrease from the present-day level (2.3 %) to 1.19% in the 'best case' future scenario (GCAM-RCP4.5 SSP1).

## Discussion

Our Ebola system-dynamics model, in the absence of case data, is able to recapitulate the known endemic area of EVD accurately,

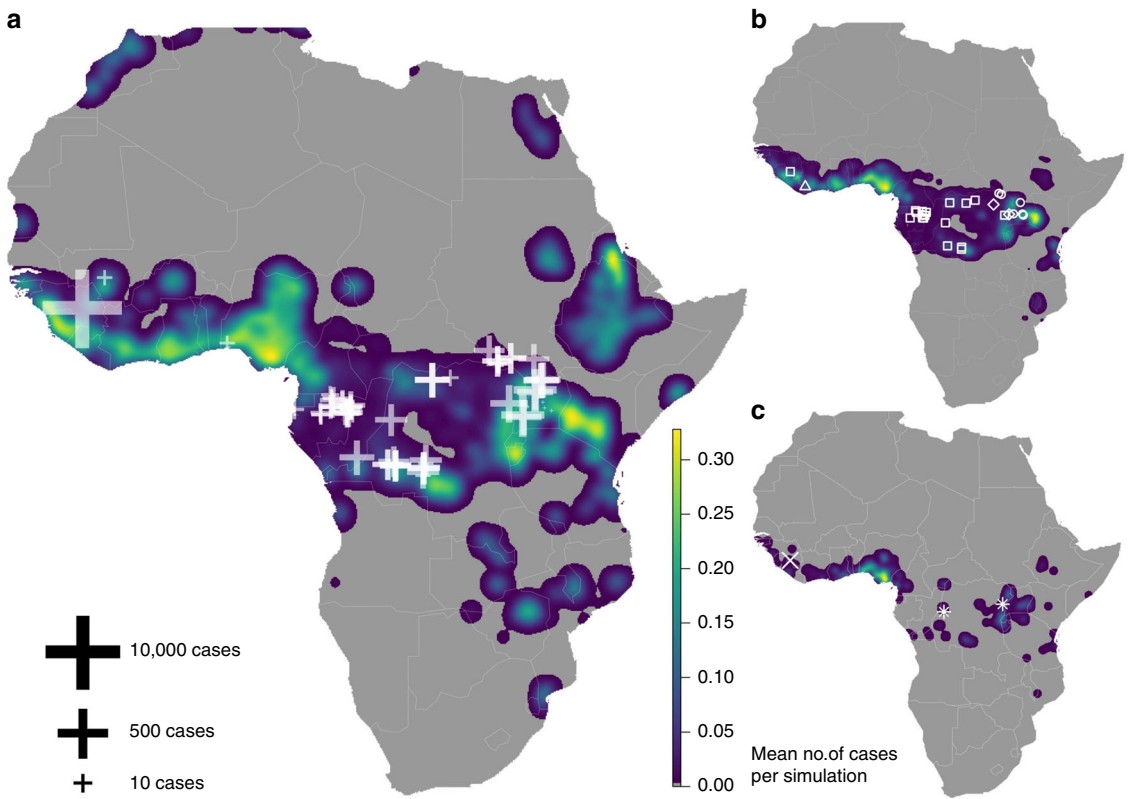

**Fig. 3** Present day risk of EVD cases caused by Zaire Ebola virus (EBOV) from EMM simulations. Maps represent the mean number of EVD-EBOV cases between zero (dark blue) and 0.3 (yellow) per grid cell (0.0416°—5.6 km at equator) across 2500, 365-day simulation runs for the present day, where (**a**) shows all cases (both index and secondary), (**b**) index cases only, and (**c**) index cases from epidemics (1500+ cases). White crosses in (**a**) represent log outbreak size. White symbols in (**b**) represent all locations of known EVD index cases from different viral strains, where circles represent Zaire (EBOV), square Sudan (SUDV), triangles Taï Forest (TAFV), and tetrahedrons Bundibugyo (BDBV). Diagonal white cross in (**c**) represents location of index case from 2014–2016 epidemics, white stars the locations of Ebola outbreaks that have occurred since the modelling was run

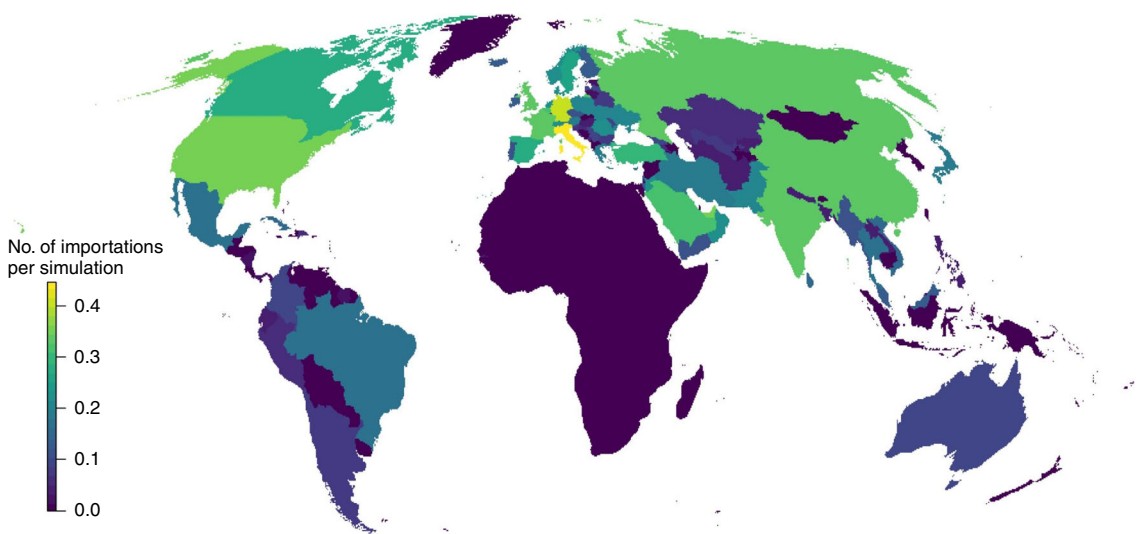

**Fig. 4** Most common country locations for importation of EBOV infected individuals. Map shows the number of importations per simulation that countries outside Africa received via airline flights. Countries with the most EBOV infected individuals imported are represented in yellow with darker green, then blue, coloured countries having proportional fewer importations and dark blue showing zero importations and the EVD endemic area. Data come from 2500 simulations of EVD outbreaks under present data climate, land-use, demographic and transportation conditions

but also suggests that other areas of the continent, such as West and East Africa are also potentially at risk. Despite there being only two known spill-overs, there is a paucity of information to discount EVD risk in West Africa, and given the lack of data to

quantify the likelihood of the 2014–2016 West African outbreak reoccurring, the risk of an outbreak in this region should be considered valid. These findings support previous work that has suggested that several countries in Africa could be at risk despite

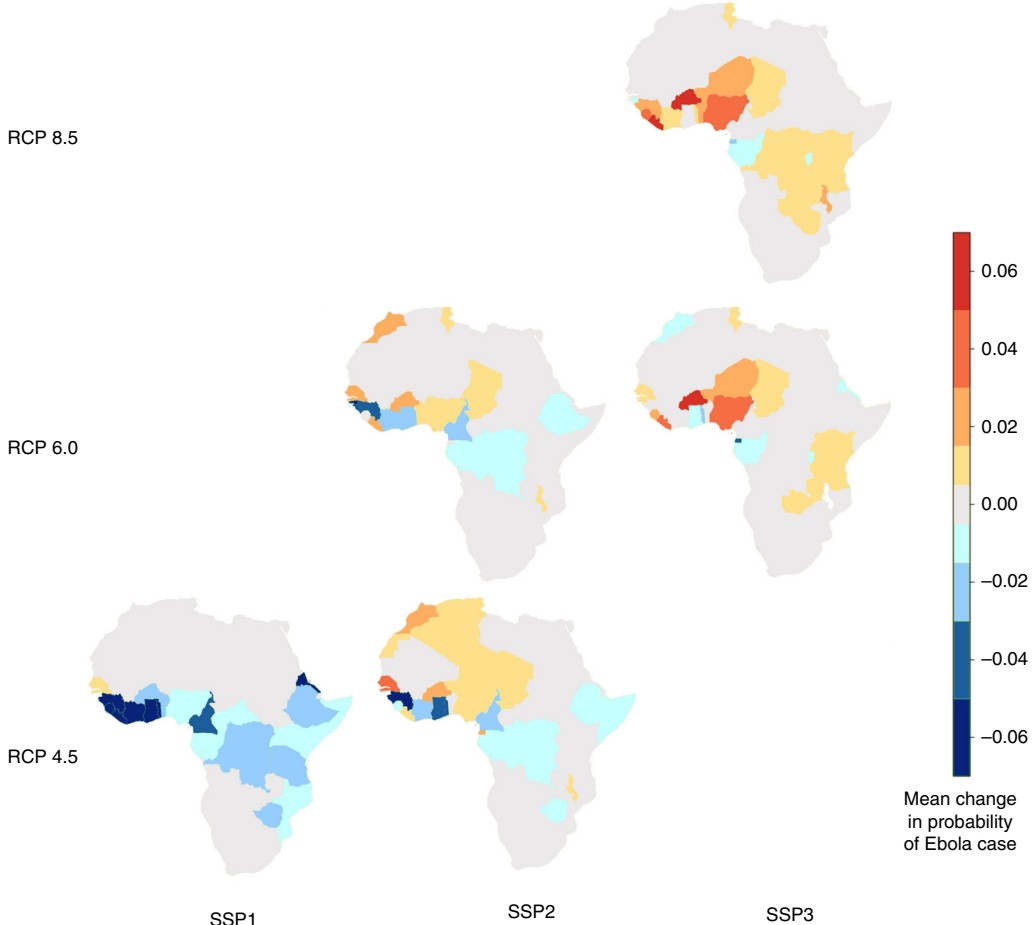

**Fig. 5** Change in future risk of EVD cases caused by Zaire Ebola virus (EBOV) for 2070. Maps represent mean change in per grid cell (0.0416°—5.6 km at equator) EVD case probability from zero (yellow) to −0.06 (green) and 0.06 (red), aggregated at the country level with data from EMM simulations for 2070. Rows and columns show all reasonable combinations of the different scenarios of global change (GCAM-RCP4.5, AIM-RCP6.0, MESSAGE-RCP8.5 and SSP1 to 3)

having experienced no known cases[16,40,41]. That our results here differ from previous case-driven analyses is unsurprising given minimal case data available to build correlation-based models and common problems associated with small data sets, such as model instability, where many differently specified models have similar likelihoods, and biases caused by the limited geographic coverage of case reports[42]. Indeed, the disagreement, for instance, we see between the observed data and simulation outputs, in terms of the frequency of small infection events, could also be due to a minimum detection threshold for outbreak size, especially given very high numbers of fevers being misdiagnosed as malaria[43]. While vaccinations and exhaustive health care efforts[44] have been effective to contain recent outbreaks, the sporadic spatial and temporal nature of spill-over events has meant it is currently unclear where preventative health care infrastructure should be best targeted. Our risk surfaces can be useful here, for instance, as we show that some parts of the endemic area have a higher chance of turning into very large outbreaks (Congo, Uganda, southern parts of West Africa). Also, the scale of our simulations is very flexible and, therefore, our framework could be employed adaptively in real outbreak situations. For each outbreak, for instance, we can simulate the expected spread of the disease and as vaccinations occur it would be possible to update the spatial inputs and rerun the model. Future work could, therefore, focus on making our model an available tool for health care professionals.

We identify Nigeria (but also Ghana, Kenya and Rwanda) as not only a key area for epidemics to be initiated, but also an area with potential for many small outbreaks, somewhat contradicting analyses based on current case data[16,40,45]. Our finding could indicate that our underlying model has not correctly balanced the impact of healthcare infrastructure on disease spread, that there are regional behavioural barriers to infection, or that there are strong regional differences in effective contact rates between both humans and hosts. Ebola outbreaks are rare, but potentially high-impact events, and our experience of endemic health care systems being challenged by EVD cases is, as yet, limited to a handful of data points, meaning it is unclear what the true relationship is between health investment and EVD risk reduction. For Nigeria, therefore, though we may be underestimating the role of the health infrastructure[46], given just a single replication of Ebola importation it would be unwise to assume the same outcome every time, especially given the thousands of different potential transport routes for infected people into Nigeria. An extreme discontinuity in the spatial distribution of Ebola virus seroprevalence or pathogenicity of competing strains in Nigeria, and surrounding countries, could explain the observed lower-risk pattern, but it would be prudent to prove this is the case using extensive bio-molecular surveys. Certainly, the seroprevalence of Ebola virus in wildlife species is a key missing dataset that could substantially aid future understanding of this disease. Until these additional factors are explicitly tested, we believe the high human

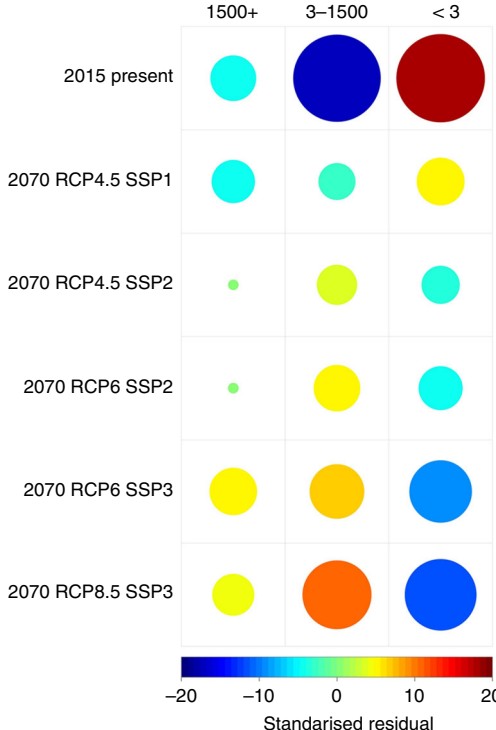

**Fig. 6** Comparison of 2070 EMM simulation scenarios by EVD-EBOV final outbreak size. Circles represents standardized residuals from a chi-squared test of association ($\chi = 466.27$, $df = 10$, $p < 0.001$) between simulation scenario and final outbreak size category. More orange/red colours show a greater than expected (vs. randomly allocated) number of outbreaks for any given combination of scenario and final outbreak size, with more blue colours representing fewer than expected outbreaks. Size of circle also indicates the overall quantity different to expected, with large circles contributing more to the overall chi-value compare to smaller circles

density and known presence of putative wildlife hosts means that all the at-risk areas we present here should remain as candidate locations for outbreaks. As an illustrative example, we recover Guinea as being at high risk of large outbreaks but if our outputs were, for instance, validated against data prior to 2013 it would appear a false prediction.

There is a similar lack of information to fully understand the true distribution of global EVD importation risk, as there have only been a handful of documented importations observed so far. The pattern we report is effectively a recreation of the shape of airline network when flying from locations that commonly experience outbreaks. Any errors made in delineation of the dominant areas that experience outbreaks will, therefore, be propagated into this prediction. Future EVD importation risk will be an emergent property of the interactions between areas of Africa that are likely to experience the disease, the healthcare response in terms of treatment and containment, and the shape and size of the airline network at that time. Future research could, using our approach, model the likelihood of index cases occurring in different parts of Africa managing to reach every country given different sets of healthcare interventions.

To further improve our model, we would need to better understand the spatial variation in other key disease transmission parameters. For instance, bush-meat hunting is an important process by which human populations come into contact with large bats[47] and the spatial variation in bush-meat extraction is likely a component of spill over variation. Little is known, however, about bush-meat hunting outside a few specific studies, but

there is potential to use spatial interpolation techniques to make reasonable predictions in unsampled areas. Our model does not incorporate this data or test its impact and, similarly due to lack of data resources, we do not use information about local differences in funeral practices. Hospital compartments are thought to be useful to understand quarantine and super-spreading events, but there is very limited data on the quality and geographic reach of small and temporary-response health clinics. Some other important behavioural trends are not captured in our model, such as the post-outbreak behavioural reactions of human populations e.g. mass migration away from affected regions, or, avoidance of clinics and treatment centres due to transmission fears. Recent findings regarding the persistence of Ebola virus in semen of convalescent men may also help explain the intermittent spatio-temporal patterns of infections in endemic areas[48,49]. Future work incorporating such data, may further improve the spatial resolution and accuracy of risk estimates.

Looking to understand the possible current direction of changes to Ebola burden over time, we project our models to 2070 and show that changes to SSP scenarios, which control levels of poverty and human population size in our models, will likely have a greater impact than climate and land-use change (here mediated via RCP scenarios). This is not surprising, as poverty reduction increases the presumed EVD-EBOV healthcare response in our simulations, and many of the countries in the endemic region are expected to have substantial reductions in poverty levels by 2070[36]. Similarly, contact rates in our simulation (both between humans and between humans and wildlife) depend linearly on human population growth, whereas climate change increases EVD-EBOV cases through more complex interactions. Species distribution models indicate that the presumed wildlife hosts prefer warm and wet conditions (Supplementary Figs. 2, 3), which are expected to increase in these regions according to the HADGem2-AO climate model[37] (Supplementary Fig. 6). This model, when compared to the predicted host change from 32 other models represents the 'middle ground' in terms of host spatial occurrence change (Supplementary Fig. 7), but future work should further examine how climate model choice impacts future predictions. This expansion of the optimal conditions for presumed the wildlife host species effectively increases the at-risk human population by including more of the northern, eastern and southern areas of Africa (Fig. 3a). Predicted future anthropogenic land-use changes, however, reduces the optimal wildlife host habitat, thereby reducing human-wildlife interactions. In terms of global socio-economic development, EVD cases numbers would appear to respond best to efforts in slowing the rate of population growth and increasing socio-economic development while mitigating climate change, such that global change most closely tracks the GCAM-RCP4.5 SSP1 scenario (High Climate Mitigation + Sustainable Development). Global binding commitments to reducing climate change may act to slow the effects, but evidence[50] suggests a wholesale change is unlikely. Efforts to decreases poverty in Central and Western Africa with a concomitant increase in healthcare resources, therefore, would appear to be the most realistic approach to reducing future EVD disease risk globally.

Our approach demonstrates not only an important framework for understanding Ebola but also for other diseases, given our modelling framework is disease agnostic and flexible in terms of compartment model structure, transmission location and scale. Analysing diseases singly, and within just an epidemiological or socio-economic framework, cannot be an effective approach for policy making at a large geopolitical scale, particularly in regions with multi-disease burden and limited healthcare resources. Net disease risk patterns, when summed across a wide variety of zoonoses, will be an emergent property of the distribution of very

different wildlife host species and their respective responses to increasing anthropogenic land-use conversion and climate change. Any lack of data in the short-term does not reduce the obvious importance of understanding future disease trends. Interdisciplinary methods, such as ours, establish a first heuristic step on a pathway towards building tools to devise future-proof intervention measures aimed at reducing overall future disease burden.

## Methods

**Analysis outline.** Focusing on the Zaire Ebola virus (EBOV), we use a stochastic epidemiological compartmental model[29], to simulate both pathogen spill-over and subsequent human-to-human disease transmission (Fig. 2). Within grid cells (0.0416°—5.6 km at equator) covering continental Africa, we used a Susceptible, Exposed, Infectious, Funeral and Removed (SEIFR) EVD-EBOV disease compartmental model following[13,19,23] to estimate the number of individuals per compartment, in each time step $t$, for present day bioclimatic, land use and demographic conditions. Although some previous compartmental models for EBOV have included a Hospital compartment[51], adding this complexity was not feasible over large and poorly known geographical areas. Without knowing more about the spatial variation in health seeking behaviour, exactly which grid cells contain clinics, and the variation of healthcare resources in these clinics, adding in this compartment would not likely significantly improve our model's ability to predict the progression of outbreaks. Furthermore, hospital interventions had the least impact controlling EVD outbreaks in a recent meta-analysis (24). Grid cell size was chosen as the highest resolution at which computation was feasible while being able to use a non-stochastic human movement model to approximate contacts per cell (see details below). All mapping and analyses were carried out in R v.3.2.2[52]. Each stage of the EMM simulation is discussed in more detail below:

**Stage 1: SEIFR compartmental model within grid cells**. We used starting EBOV transmission characteristics of incubation time = 7 days, onset of symptoms to resolution = 9.6 days, maximum case fatality rate for very low income countries (CFR) $\sigma = 0.78$, and burial time = 2 days[23] to parameterize the SEIFR compartmental model to determine transition rates $\alpha$ (between Exposed to Infectious compartments), $\gamma_\sigma$ (Infectious to Funeral), $\gamma_{1-\sigma}$ (Infectious to Removed), and $\gamma_F$ (Funeral to Removed) (Fig. 2). To incorporate sensitivity around these transmission parameters, we allowed values to vary for each simulation run by sampling from a Gaussian distribution where the mean was their initial value and the standard deviation was fifth of the mean, to give a reasonable spread of values. For each time step $t$, the number of individuals moving between all compartments was estimated by drawing randomly from a binomial distribution (Supplementary Equation 1), parameterized using the respective compartmental transition rates. Transition rates for compartments were assumed to be the same in all grid cells except for the transition between Susceptible to Exposed. The per grid cell Susceptible to Exposed transition rates were determined by the force of zoonotic infection $\lambda_z$, and the force of infection $\lambda$ (Fig. 2) and these were calculated as follows:

(a) Force of Zoonotic Infection, $\lambda_z$. The force of infection for zoonotic transmission $\lambda_z$, per time step $t$, was estimated as the product of the probability of host presence $H$, and spill-over rate $\kappa$ (Supplementary Equation 2). Without any evidence to the contrary[15,53], we parameterized $H$ by calculating the spatial probability of the presence of the most likely EBOV reservoir host species based on available data (Old World fruit bat species *Epomophorus gambianus gambianus*, *Epomops franqueti*, *Hypsignathus monstrosus*, and *Rousettus aegyptiacus* see Supplementary Table 1) within each grid cell across the African continent using species distribution models (SDMs)[54] and assuming constant pathogen prevalence. We also calculated the spatial probability of the presence of other species which are known to provide an alternative route of infection, but likely do not act as reservoirs (*Gorilla spp.*, *Pan spp.*, and *Cephalophus spp.*)[12]. SDMs for each species were inferred using boosted regression trees (BRT) using distribution data from the Global Biodiversity Information Facility (GBIF)[55] and 11 present day bioclimatic and land use variables (Supplementary Table 2). Data with coarse scale GBIF spatial coordinates (decimal degree coordinates with less than four decimal places) were filtered out of the analysis. To reduce spatial autocorrelation and duplicate records, any records that co-occurred in the same grid cell were removed. Lastly, GBIF records older than 1990 were discarded to ensure samples more closely matched the current landscapes. BRT tree complexity was set at 5 reflecting the suggested value and the learning rate was adjusted until >1000 trees were selected[56]. A total of 25 models were estimated for each species using four-fifths of the distribution data as a training dataset and one fifth as a testing dataset, chosen randomly for each model. Those with the highest predictive ability (high area under operating curve, AUC—all models AUC > 0.9) were selected as the best model for each species (Supplementary Fig. 2). The most important spatial variables determining distributions across the different reservoir host species were BIO7 Temperature Annual Range, BIO13 Precipitation of Wettest Month, BIO2 Mean Diurnal Temperature Range and Land Use-Land Cover (Supplementary Fig. 3). The outputs from all putative reservoir (bat) species were combined into a

single value representing the probability of any reservoir species being present and a similar approach was taken for the non-reservoir host species. First, we assume that complete mixing occurs within each grid cell and that if secondary host species are present in a cell they meet the presumed primary hosts. Then we calculate one minus the probability of each host presence, multiplied across all species, giving us the probability of there being no species present in any given cell. One minus this value gives the probability of at least one species being present. We assumed that bats were the default, fundamental component for transmission (i.e., acting as the 'working' reservoir) and, given complete mixing in each cell, apes and duikers were a secondary, less common route of human infection. Since roughly a third of index cases have been attributed to ape/duiker host spill-overs[10], in cells with both groups we down-weighted the probability of the ape/duiker occurrence by two thirds and reservoir occurrence by one third when combining the two lots of layers together, though, we note, due to the similar habitat requirements a very high majority of the cells containing duikers and apes also had high probability for bats so the precise value here will have a limited impact. The final resulting probability was bounded by zero and one. Additionally, as EBOV presence in non-reservoir host species is impossible without the presence of reservoir hosts, cells with a reservoir host probability of zero were given a value of zero irrespective of the non-reservoir host score. For computational simplicity, we assume that all human individuals have equal chance of exposure to infected host species. The initial value used for spill-over rate $\kappa$, per time step $t$, was estimated from the number of historic outbreaks $O$ (defined here as distinct clusters of cases) taken from empirical EBOV outbreak data[12], and the number of historically susceptible individuals $S_h$ (inferred from human population estimates from 1976 to 2015 from[36] (see Supplementary Equation 3). During each simulation run, $\kappa$ was allowed to vary using the same method as for the compartmental transmission parameters above.

(b) Force of Infection, $\lambda$. The force of infection for human-to-human transmission $\lambda$ per time step $t$, was estimated as the product of the effective contact rate $\beta$, and the number of individuals that can transmit the disease in each relevant compartment ('Infectious' and 'Funeral') per grid cell (Supplementary Equation 4). We assumed that $\beta$ for the Infectious and Funeral compartments was equivalent, due to the simplicity of the movement model used as we would not be able to reasonably differentiate between the two compartments and maintain computational efficiency. Indeed, with the contact rates of moving individuals from the Infectious compartment being offset by aggregations of individuals at funerals it is not clear if there would be a large difference if approximated using a gas model. We estimated the effective contact rate $\beta$, as the basic reproduction number $R_0$ divided by the product of the total number of individuals $N$, and infectious duration $D$ (the sum of Infectious and Funeral compartment time, 11 days taken from[23]). As a starting value for $R_0$ we used a value of 1.7[57] and this was allowed to vary per simulation run using the same method as for the compartmental transmission parameters above. As per previous research[29], we incorporated spatial variance in contact rates among grid cells using a weighting factor $m$, whereby the effective contact rate in grid cells with greater than expected contact rates was increased and decreased where fewer contacts were predicted (Supplementary Equation 5). We estimated $m$ by creating an ideal free gas model of human movement within each grid cell and approximated collision frequency per person per day, using the following: the total individuals in each grid cell (estimated from Gridded Population of the World v3[58], an individual interaction sphere of radius 0.5 m, and the per person, daily walking distances in metres $v\Delta t$, where $v$ is walking velocity, and $\Delta t$ equals time period (Supplementary Equation 6). To capture geographic variation in human movement patterns, each grid cell was assigned a value for per person daily walking distance, based on the empirical relationship between daily walking distances and per person per country Gross Domestic Product (measured as Purchasing Power Parity from[36]) (Supplementary Table 3). As the availability of mass transit as an alternative to walking tends to be centrally controlled, we assumed that grid cells in each country had the same value.

Under observed conditions, the effective reproduction number $R_e$ decays over time as both efforts are made to control disease spread and the pool of susceptible reduces, which results in $R_0$ being equal to $R_e$ only when time $t$ is zero. Therefore, to calculate effective contact rate $\beta$, we allowed $R_e$ to decay per time step $t$ (Supplementary Equations 7, 8 and 9). However, countries that can invest more in health infrastructure (e.g., barrier nursing, surveillance) should see a more rapid reduction in $R_e$ over time compared to countries that do not have such infrastructure and also a concomitantly, a decrease in CFR. Therefore, we derived an empirical estimate of the relationship between wealth (measured using GDP-PPP per capita) and both the relative rate of decay of $R_e$ over time (Supplementary Equation 10) and CFR (Supplementary Equation 11). Using a spatially disaggregated poverty data layer[59] we weighted the per grid cell per time step $R_e$ reduction and CFR accordingly to the values in each grid cell. While we found the relationship between wealth and both $R_e$ and CFR reduction over time to be best described using curves with exponents of −0.08 and −0.02, respectively, this was inferred using relatively few data points (Supplementary Table 4). In our simulation runs, therefore, we allowed these exponents to vary similarly to the parameters above, to allow either more linear declines or deeper curves to best estimate the true impact of this relationship.

**Stage 2: SEIFR compartmental model between grid cells.** We allowed those individuals that had contracted EBOV to travel between grid cells (specifically

individuals in Exposed and Infectious but not Funeral compartments) (Fig. 2), but assumed for simplicity that the overall net movement of susceptible individuals between cells was zero. As previously supported with empirical data, we employed a movement model that was weighted by both geographic distance and human density[30,32], and was also geographically constrained to known transportation routes. The transmission rate $\varepsilon$, of individuals between target compartments of different grid cells was estimated by two different methods: between grid cells along road networks $\varepsilon_r$, and along flight routes $\varepsilon_f$. We sampled randomly, from a binomial distribution, the number of travellers per grid cell and time step $t$ (Supplementary Equation 1) with the probability of travel by road per day $\varepsilon_r$, being proportional to the distance to the nearest road using the Global Roads Open Access Data Set (Global Roads Open Access Data Set from[60]). Global roads dataset contains in total 585413 routes from tracks to multi-lane highways and has been extensively validated for Africa[60]. We allowed travellers to move freely (agnostic to any particular transportation method or country boundary) across the continent up to 10 road junctions in any directionalong the road network starting from the centroid of the target cell (Global Roads Open Access Data Set from[60]), giving a potential of up to 500 km of linear travel per time step. Each proposed travel end point was given an individual probability from the daily distance travelled probability curve from Fig. 2f of ref. [61], which is derived from transport data and validated against mobile phone data. For air travel, we set the potential pool of travellers as the individuals in grid cells containing airports across the world from Open Flights Airport Database[62], plus all the Exposed individuals in the 8 grid cells surrounding each airport grid cell. We sampled randomly from a binomial distribution the number of travellers per grid cell and time step $t$ (Supplementary Equation 1) with the probability of travel by air per day $\varepsilon_f$ being proportional to the total number of flights per day divided by the population of that country[36]. We allowed travellers to move up to 2 edges on the current airline routes from airport origin using[62]. This approximates a traveller taking either a one or two-legged journey. Final destinations were sampled at random, based on all potential air routes having equal priority, but in most cases potential destinations were located nearby, which by default meant that more distance travel was less likely than travel to a nearby location. We decided to keep each edge on the network as equal likelihood due to a lack of comprehensive and detailed information that we could find on passenger numbers at the time of modelling. For both road and air travellers, individuals were then added to the correct compartment of their final destination in the new grid cell and removed from the same compartment of the original source grid cell.

**Stage 3: Impact of future anthropogenic change**. (a) Future force of zoonotic infection $\lambda_z$. We recalculated values of the force of zoonotic infection $\lambda_z$, by estimating the probability of EBOV host presence, $H_{2070}$ under several different future integrated scenarios that incorporate projections of bioclimatic and land use variables (Supplementary Table 2). Estimates of bioclimatic variables for 2070 were based on the HADGem2-AO climate model[37] under three Representative Concentration Pathways: CAM-RCP4.5, AIM-RCP6.0, and MESSAGE-RCP8.5[39,63,64]. These different options were, specifically: (i) GCAM-RCP 4.5 (High Climate Mitigation)—stabilization scenario in which total radiative forcing is stabilized shortly after 2100, (ii) AIM-RCP 6.0 (Low Climate Mitigation)—stabilization scenario in which total radiative forcing is stabilized shortly after 2100, without overshoot, by the application of a range of technologies and strategies for reducing greenhouse gas emissions (iii) MESSAGE-RCP8.5 (Business as Usual Emissions)—worsening scenarios with ongoing increasing, unchecked, greenhouse gas emissions over time, leading to high greenhouse gas concentration levels. Although we only used a single overall climate model (HADGem2-AO) due to computational constraints, this model offers good agreement with other alternative models[65] and a 'middle of the road' option in terms of realised changes to future host distributions compared to 32 other models (Supplementary Fig. 7; Supplementary Table 5). To estimate host presence probability in the future we needed to predict fine-scale future habitat data under the RCP scenarios. As only coarse land-use categorisations are currently available[66] with, for instance, a 'primary' land-use having a wide variety of possible natural habitats from arctic tundra to tropical rainforest, we therefore separately empirically estimated future land use-land cover (LULC) change using the spatiotemporal patterns contained within the MODIS land-cover time series[35]. For each grid cell, at the same grid cell resolution as set out above, we calculated the frequency of each LULC change seen in the 2001–2012 MODIS dataset for the surrounding square of 25 grid cells around each grid cell. We converted this frequency to a probability by dividing by the total possible number of changes, and based on these probabilities, we simulated yearly LULC change across the region of interest for each grid cell from 2012 until 2070, and ran this simulation 100 times to create a bank of future possible landscapes. These were then summarized into three consensus landscapes representing low (with anthropogenic changes rejected where possible), medium (by choosing the majority consensus across all 100 runs) and high anthropogenic change, (more anthropogenic changes (e.g. urban, cropland, mosaic cropland) were preferentially chosen across the landscape) and we aligned these three scenarios to SSP1, SSP2 and SPP3, respectively (for scenario details see below)[38,39].

(b) Future force of infection $\lambda$. Using predicted human demographic variables and poverty levels for 2070, we recalculated values for the force of infection $\lambda$, by estimating the number of individuals per grid cell, $n$ and effective reproduction number, $R_e$. We inferred human population estimates per grid cell for 2070 by using the Gridded Population of the World v4[58] for present day and multiplying each cell by the expected future proportional change over that time period predicted by three Shared Socio-economic Pathways: SSP1, SSP2 and SSP3. Specifically, these pathways represent: (i) SSP1 (Sustainable Development)—high regional cooperation, low population growth due high education and high GDP growth, (ii) SSP2 (Middle of the Road Development)—a 'processes as usual' scenario with ongoing levels of population growth and wealth, with medium estimates for both these by 2070, and (iii) SSP3 (Regional Rivalry Development)—regional antagonism, high population growth, unsustainable resource extraction and low economic growth. Future poverty estimates per country were similarly inferred using a spatially-disaggregated GDP layer[59] multiplied by the expected change in per country GDP over the time period as predicted by the SSP integrated scenario, with any future changes in wealth interacting with the $R_e$ parameter to affect disease epidemiology accordingly. We note that as our travel probability is defined per person, increasing future populations will see a proportional increase in the amount of both road and air travel, though with unknown patterns of future trade and travel we kept the travel network the same shape.

(c) Comparison of simulation runs. We reran the EMM simulations under 5 plausible combinations of 2070 future environmental and socio-economic scenarios of global change and greenhouse gas concentrations, matching greater greenhouse gas emissions to less progressive and less cooperative socio-economic scenarios as follows: GCAM-RCP4.5/SSP1 (High Climate Mitigation + Sustainable Development), GCAM-RCP4.5/SSP2 (High Climate Mitigation + Middle of the Road Development), AIM-RCP6.0/SSP2 (Low Climate Mitigation + Middle of the Road Development), AIM-RCP6.0/SSP3 (Low Climate Mitigation + Regional Rivalry Development), MESSAGE-RCP8.5/SSP3 (Business as Usual Emissions + Regional Rivalry Development)[38,39,64,67]. For each of the six scenarios we aimed for 2500 runs of 365 days, each day measuring the number of spill-overs, the number of secondary cases associated with each spill-over, and the geographical areas affected. This allowed us to measure likelihood of spill-overs leading to small, medium and very large outbreaks, and also to determine the geographical areas with the highest risk of experiencing cases. We also noted the destination of any flights out of Africa that contained infected people.

**Reporting summary**. Further information on research design is available in the Nature Research Reporting Summary linked to this article.

## Data availability

The datasets generated during and/or analysed during the current study are available in the figshare repository, https://doi.org/10.6084/m9.figshare.1559959

## Code availability

Code used during and/or analysed during the current study are available from the corresponding author on reasonable request.

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

## Acknowledgements

We thank P. Sivasubramaniam for technical assistance, and A.P. Jones, M. Wilson, G. Mace, M. Leach, B. Sheldon, and C. Watts for comments on previous versions of the manuscript.

## Author contributions

D.W.R. and K.E.J. designed the study, D.W.R. devised and ran analyses, D.W.R., P.M.A., A.A.C., G.L.I., L.M.M., J.L.N.W. and K.E.J. all contributed to the writing and editing the paper.

## Competing interests

The authors declare no competing interests.
