## [Peer Review File · Nature Communications]

Reviewers' Comments:

Reviewer #1:

Remarks to the Author:

Review of "Modelling impacts of global change on emergence and epidemic potential of Ebola in Africa"

Manuscript number: NATCOM-18-16450

Authors: Redding et al., 2018

Recommendation: Major revisions

General:

This study aims to model the risk of Ebola virus disease (EVD) over Africa. This is carried out for the recent context and for the future (2070) using a combination of climate change, demographics and land use scenarios. Interestingly, the authors investigate the risk in animal bat hosts and the potential for virus spill-over to human hosts. The model is validated with respect to observed outbreak data before providing future risk scenarios. The authors show that future changes in the drivers of Ebola might result in a 1.75 to 3.2 fold increase in the endemic rate of EVD outbreaks by 2070.

Overall the paper is well written and the approach is novel e.g. the authors look at global change impact (including dynamic estimates of climate, demographic and potential land use change) on EVD risk (using all cases, index cases only and index cases from epidemics as output metrics).

Consequently, I recommend this study to be published in Nature communications. I just have many minor comments that need to be addressed before acceptance of the manuscript; this is why I suggest major revisions.

Minor points:

Abstract: "human socio-economic conditions" – remove "human"

Abstract: "might result in a 1.75 to 3.2 fold increase...". Looking at the result section, I could not find these estimates (smaller numbers are discussed in the Future trajectories section; these are mainly based on suitable surface ratios; please clarify)

Abstract: "We show that trends in the underlying ... will result ... healthcare intervention". This is a modelling study and uncertainties included in future scenarios can be quite large. Thus, please remove the deterministic "will result" by "might result" given these uncertainties. You also need to mention the study region (... in Africa) as a naïve reader might think this is a global estimate.

Intro – page 3: "global health threat and economic burden"

"to better forecast zoonotic disease risk and emergence"

"with a high case fatality rate" – true but we used to think that Ebola had a 80% mortality rate based on the old DRC outbreak data, while the observed mortality rate was far lower (about 50%) during the recent west African outbreak.

Intro – page 5: "wild host ecology"

Fig 2 caption: The poverty-weighted case fatality rate for EBV is quite large (78%) while it was lower during the recent outbreak in West Africa (50ish %)

Results – present day patterns: please be more descriptive about the African countries where EVD has not been reported yet (list them all e.g. Togo, Tchad, cote d'Ivoire, etc) as this is a valuable information.

Fig 3: Can you replace the black symbols (dots) which represent the locations of known EVD index cases to white dots or light grey crosses? Black on dark blue is difficult to read

Fig 4: Map of importation risk of EBOV infected individuals. Where is the keybar / scale ? The risk ranges from yellow to red but we have no idea whatsoever about the magnitude of that variable.

Page 11 – Future trajectories: "but increased by 14.7% under ...". I calculated the surface ratios and it should be 13.9% (note that this might be related to rounding of the surface estimates). Please clarify.

Fig 5: The authors should provide difference maps e.g. deltas calculated between the future risk maps

and the baseline map shown on Fig 3 (these maps can be included in Sup. materials for example).

Discussion: "ameliorate climate change", prefer "mitigate climate change".

Discussion: "it is currently unclear where vaccination should be targeted...". Ok, but you plot risk maps so you could highlight countries at risk which were affected and countries where epidemics could occur (I guess this is the point of a risk model?)

Discussion: "vaccination estimated at \$135.90 per person".

Discussion: The largest risk seems to be simulated over Nigeria. There is a paragraph in discussion mentioning this issue. The authors need to mention that Nigeria was slightly affected by Ebola; they had it under control rapidly, thanks to their public health services. This needs to be discussed in this paragraph: <https://europepmc.org/articles/pmc4584877>

Methods: A few important things to clarify in this section:

1. The authors used the Gridded population of the world dataset v3 to obtain counts or densities of population. This dataset is available at 30arc sec (about 1km x 1km at the equator). Then they use an interaction sphere of 0.5m combined with daily walking distances to calculate geographic variation in human movement patterns. There is a spatial scale mismatch between this dataset and the approach (1km x 1km vs 0.5m – the 0.5 m radius for the whole of Africa might be super expensive computing wise as well). Please clarify, I might have been confused here

2. The climate model runs are only based on the UKMO model (Hadgem2-AO). The authors mention that this model is close to other climate models. However, changes in rainfall (and to a lesser extent temperature) can vary a lot across different climate models. As an example, for West Africa, it is difficult to draw any conclusions on the CMIP3 or CMIP5 climate model ensemble (some models predict drier conditions, other no changes, other simulate drier conditions). The authors need to mention that they did not include uncertainties related to the chosen climate model, and these can be quite large, in particular over the African continent.

Reviewer #2:

Remarks to the Author:

Major Comments

In "Modelling impacts of impacts of global change on emergence and epidemic potential of Ebola in Africa" the authors combine an epidemiological model with projected estimates of future climate, land use, and human population change to estimate the risk of both a spillover event from Ebola and the subsequent epidemic. Incorporating both of these factors in a single model is a fairly reasonable approach and it is reasonable to investigate how various socio-economic and climatic projects will impact a pathogen like Ebola. Overall, the authors provide a reasonable analysis of these factors, however four key points must be addressed to fully assess the validity of their approach and analysis.

In general, their current model results (for present day) are fairly inconsistent with known outbreaks. The authors state that underreporting prevents them from performing a validation exercise, however all diseases are underreported and without statistical support it is unclear to what extent their model framework and parameterization is valid. Finally, the future projections are fairly inconsistent amongst themselves. Often times these results contradict one another, or provide a very wide range of estimates. As a result, it is unclear what general conclusions can be drawn from these results.

1. The model should be validated against existing data.

a. It is unfair to use underreporting as justification to not perform a proper validation exercise. All diseases are underreported, however these data are still incredibly valuable and should be used to calibrate their model.

b. The only comparison of their model results with data are too casual and not backed up with a

proper statistical analysis. This should be corrected in future versions of the manuscript.

c. Their model predicts there is a strong spillover and subsequent outbreak risk in Nigeria and Ghana, two countries that the authors acknowledge have not reported Ebola outbreaks. The authors should provide justification for why their model produces these results. Given that they do not perform a proper validation exercise, this result makes the reader question the validity of their model and all subsequent results. Underreporting is not a strong enough justification for these results, particularly given that Nigeria and Ghana have two of the strongest healthcare systems in West Africa. These results imply that their model is not well-structured or parameterized to accurately capture Ebola dynamics.

d. The result that the spillover risk is greater than the known area is unsurprising and the authors need to provide additional evidence that they are capturing known spillover areas and why their model is overestimating in these specific areas.

e. The importation risk is highest in countries that have not experienced many importations of Ebola, even during the most recent large outbreak. Can the authors provide justification and further support for why their model is producing these unexpected and inconsistent results?

2. The authors should assess the public health impact of these results.

a. In the introduction, the authors mention that this could be used to design adaptive vaccination programs, however they do not expand upon this in the main text. What are the implications of these results?

b. Also, vaccination in response to an Ebola outbreak will likely be reactive. Can the authors provide how their results will assist in this type of strategy?

c. It is not an actionable, helpful recommendation to reduce the population size. Could the authors provide more relevant recommendations?

3. The projection models require clarification.

a. Given the interdisciplinary nature of the journal, the authors need to provide the reader with additional information about the various projection models used. Although these models may be very common in the climate change literature, they are likely unfamiliar to this audience. It would also be helpful if the authors created shorthand for each model (aside from the abridged name) to assist the reader in recalling the general assumptions/framework of the different model projections.

4. The results from the different projection models do not agree. To what extent can these results be used to draw general conclusions?

a. Often the results from the various climatic and socio-economic projection models produce contradictory results with a wide range of estimates (ex: decreases by 47% in some instances, but increases by 34% in other models). Given the inconsistent results, what is the main take away from these results?

b. In some instances, spillovers increase and in other instances they decrease. Given this, what can actually be interrupted from these results?

c. The projection models require additional clarification. For example, the authors use three different socio-economic development scenarios but do not provide enough detail about the differences between these scenarios.

Minor Comments

1. Why do the authors project forward to 2070?

2. During the Ebola outbreak, health clinics that were specifically designed to treat the disease contributed to its control – these are inherently related to hospital infrastructure and health care

treatment ability and should not be disregarded.

3. What is the motivation for the current grid scale? The authors perform a very detailed spatial analysis, but do not utilize these results. And if their overall results do not abide to current wisdom and are not calibrated to actual data, then what is the value of having such a fine resolution?

4. How is healthcare incorporated into their model? Presumably it is incorporated in some way with the socio-economic projections, however it is currently unclear.

5. Their model is fairly Ebola specific and it is unfair to state that it will be useful for other diseases.

6. They produced a stochastic model, but their results do not appear to include a lot of stochasticity. How was stochasticity incorporated?

7. On Page 4, the authors state that mechanistic models rarely incorporate spatial heterogeneous ecological and environmental information. This is untrue, particularly for vector-borne disease models.

8. In Figure 3, what is the unit of the values? For example, the values go up to 0.01, but is this a percentage?

9. The authors consider an epidemic if there is at least 1500+ cases, what is their epidemiological (not purely statistical) justification? How do their results change when this arbitrary value is varied?

10. In Figure 3, you cannot clearly see the black points. This should be changed in a revised version and statistical fitting should be done and included as part of this figure.

11. In Figure 4, the resolution should be improved and a scale bar should be provided. For example, what is the difference in risk between red and orange?

12. In Figure 5, if the proportion of time is equal to 0 (shown in dark blue), why is it included in the area projections? And corresponding figure?

13. In Figure 6, can the authors provide an interpretation of these values? Is it greater than expected, as in 20x or a 20% increase? What is the significance value? What is the statistical test?

14. What is the relationship between wealth and R_e over time?

15. What is the airline data used in the analysis? Could the actual destination data and weights based on the frequency of flights and/or passenger sizes be used?

16. This analysis assumes that the population will increase, but the road and airline networks will not. Is this reasonable?

17. Why is the model run for a single year?

Reviewer #3:

Remarks to the Author:

Review of Redding et al., Nature Communications

General

The contribution from Redding et al., is undoubtedly an impressive and multifaceted piece of work. It is very ambitious in its aims of integrating mechanistic with correlative modeling approaches for the analysis of Ebola virus disease (EVD) outbreaks. The paper contributes predictions of present day risk for EVD outbreaks and is then combined with scenario-based forecasting to evaluate the way that environmental and social changes expected to 2070 may influence this risk. This could be very useful for imagining management responses and decision making on preventive measures. Importantly, the contribution attempts to close a number of significant gaps in our knowledge regarding the connectivity and inter-dependence of a large range of socio-ecological processes that together contribute to EVD's risk profile.

The approach is nevertheless rather overwhelming (as are many systems-based models) and it seems as if any number of the individual components of the model could benefit from separate peer review from appropriate experts (or even separate papers) in order to really scrutinize how the model is operating together, how reasonable this is, and how this links or extends to and goes beyond existing work and expertise on EVD. There is so much squeezed into this ms to digest.

This could be regarded as both a positive and a negative – positive in the sense that there are certainly a number of fantastic and novel advances here and this is an appealing and novel integration of ecological and epidemiological approaches that builds on from the authors' previous work developing such methods. On the other hand, lack of validation seems to be a major hurdle (as it is for many emerging disease problems) and there is very limited comparison with and referral to existing studies, particularly Ebola studies that make similar risk predictions, and the discussion is exceedingly short given the content of the ms. This makes it pretty difficult to know how well the model is really performing in its primary task, how much it departs from existing insights, and therefore how much of it will ultimately prove beneficial for EVD management over and above what is already available.

Nevertheless, despite these reservations and a few other more specific issues (see below), on balance this is a very interesting and compelling piece of work, and seems likely to stimulate a lot of discussion and thought in the field.

Specific

No line numbers in ms for review. Pages indicated below

P4 – “One downside is that mechanistic models rarely incorporate spatially heterogeneous ecological and environmental information, such as the known high variance of bat abundance and pathogen sero-prevalence across widespread individuals.” – it's not clear how/whether the correlative components used address these two example issues.

P4 – “but no studies that we are aware of have considered whole- system analyses for major epidemic zoonoses, such as Ebola.” – what about other diseases? Not entirely clear from this statement how novel systems analyses are for infectious diseases more broadly.

P4 – “Like other rare or poorly- sampled diseases, Ebola suffers from limited data availability, meaning

pattern-finding, correlative analytical techniques are at a disadvantage” – for demonstration of this novel systems approach, especially extending to scenario-based projections, is EVD really the best test case for this reason? Although I completely agree with the premise, there seem to be two somewhat competing agendas here: demonstration of the novel integrative method, and improving models for EVD risk assessment. The former may be better served by looking at a disease with greater availability of data for adequate validation. The latter arguably requires the former to be maximally convincing for data poor applications. The projections make a third key deliverable of the ms, but this is arguably undermined until both the first and second points are addressed. It’s exciting, but just a massive amount of uncertainty to squeeze into a single paper.

P4 - The authors say that there is a “trade-off between increasing human populations and loss of reservoir host species through anthropogenic land-use conversion”. However, the presence of fruit-bat species suspected to be linked to the Ebola virus, such as *Eidolon helvum* (Haiman et al. 2010, PLOS ONE 5: e11978, though not considered by the authors) seem to have responded positively to land-use conversion from forest to agriculture, and even keep large urban colonies (Peel et al. 2017, *Acta Chiropterologica* 19: 77-92). In addition, two recent papers in *Scientific Reports* document links between recent deforestation and increases in EVD risk (Rulli et al., 2017; Olivero et al., 2017).

P8 - “the at-risk area for EBOV-EVD is much larger than the areas known to have experienced disease outbreaks thus far”– at the risk of appearing self-promoting, a number of other studies have come to the same conclusion and could be referred to here and results compared/contrasted (e.g.,: Pigott et al. 2016, *eLIFE* 5:e16412; Olivero et al. 2017, *Mammal Review* 47:24-37; Murray et al. 2015, *PNAS* 112:12746-12751;). That countries or parts of countries that could be at risk that have never experienced Ebola outbreaks and therefore may be insufficiently prepared is an important message coming through from all of the above studies, including this one.

P8 – “Our risk map also identified areas that are endemic for the other EVD strains, likely due to similar transmission pathways and reservoir host characteristics (Fig. 3A).” Does this indicate a lack of specificity for the target organism due to e.g., lack of specific knowledge of hosts/reservoirs or some other factors among strains? If so, perhaps there is an argument for including information on the other strains too in the model building and validation processes?

P9 – “High risk of Ebola case importation using the current network of airline flights was seen in China, Russia, India, the United States as well as many high-income European countries (Fig. 4).” How do these results compare with observations of imported cases globally so far?

Fig. 3 B - black symbols are difficult to see. Circles, triangles and tetrahedrons are not well resolved.

Fig 4 – what defines the EVD endemic zone? North Africa but not e.g., South Africa is indicated, while Fig 3 shows most of North Africa as being unsuitable while at least a bit of South Africa being suitable?

Fig S3 – unclear how to interpret the violin plot, it seems like a qualitative validation step? Can more formal validation tests be conducted to help with interpretation regarding model performance benchmarked against observations?

P16 - Nigeria is identified here as a key area (in fact the most important area) for epidemics to be initiated and with potential for many small outbreaks. This is in contrast to previous studies (e.g., Peterson et al. 2004, *Emerging Infectious Diseases* 10: 40-47; Pigott et al. 2016, *eLIFE* 5:e16412; and Olivero et al. (2017, *Mammal Review* 47:24-37). Again, it would be beneficial to discuss the results in light of predictions from previous studies.

P19 - The outputs from models for different species (first reservoirs, then non-reservoirs, and finally reservoirs with non-reservoirs) are combined. These outputs seem to be probability values. The presence probability of a species is influenced by environmental conditions, but also by the prevalence of its distribution within the study area. Consequently, probability values for species with different prevalence are not comparable. It is thus important to include here an explanation of how these models were combined.

P19 - When combining models for different species, the authors "down-weight the probability of non-reservoir occurrence by two thirds and reservoir occurrence by one third". This decision is based on the fact that "only a third of index cases were attributed to non-reservoir host spill-overs". However, none of the index cases recorded up to now has been proved to be a result of contacts with species presumed to be the reservoir (i.e. bats). This justification may thus require reformulation.

P20 - "We assumed that β for the Infectious and Funeral compartments was equivalent, due to the contact rates of moving individuals in the Infectious compartment being offset by large aggregations of individuals at funerals." This seems like quite an assumption. It was not clear if this was addressed through sensitivity analyses.

P23 - "...with the probability of travel by air per day ϵf , being proportional to the total number of flights per day divided by the population of that country". Could this assumption overestimate importations, since economic restrictions would make it difficult to afford flights to USA/Europe etc.? Maybe an overestimate to assume that the whole population is 'available' to travel?

P23 - The method of constructing projected land-use cover and change seems pragmatic. However, this may be perceived as a major oversimplification for those in the trade of land-use change modeling, and both its accuracy and integration with other elements of the projections do not appear to have been tested/validated. As the authors acknowledge, at coarser resolutions, there are existing products available that provide harmonized land use change projections. It's not clear what the objection to using the existing harmonized dataset really is from the description, it could be the resolution on number of land cover classes or the spatial resolution. In either case, more description is required, and it would be informative to do some basic benchmark testing of the bespoke analysis against these or other similar products (e.g., via downscaling) in order to get some basic information about its robustness. Furthermore, a robust (or even indicative) 5x5km land-use change product for the region and period indicated would be a novel contribution in itself and of wide potential application and interest. However, it might also be worth considering the value of sticking to the 5x5km resolution in the first place if this can't reasonably be represented for key datasets in the analysis.

Kris Murray, Imperial College London
Jesus Olivero, University of Malaga

NATCOM-18-16450 (Redding et al.)

Responses to referees' comments

We thank the editor and referees for their valuable and constructive comments which have very significantly improved the manuscript. Please see our detailed responses in **red** below. Line numbers refer to the revised manuscript (in simple mark-up view only) unless otherwise stated. Citations to publications used to address comments are listed at the end of the current document.

Reviewer #1 Comments

Overall the paper is well written and the approach is novel e.g. the authors look at global change impact (including dynamic estimates of climate, demographic and potential land use change) on EVD risk (using all cases, index cases only and index cases from epidemics as output metrics). Consequently, I recommend this study to be published in Nature communications.

We thank the reviewer for the positive comments and the recognition of the role of this type of modelling in understanding Ebola outbreaks and generating future research topics.

Ref1 Comment1 -

Minor points:

Abstract: “human socio-economic conditions” – remove “human”

We have removed this word (line 23).

Abstract: “might result in a 1.75 to 3.2 fold increase...”. Looking at the result section, I could not find these estimates (smaller numbers are discussed in the Future trajectories section; these are mainly based on suitable surface ratios; please clarify)

We have clarified this part and made a clearer reference to the spill over rates reported in the results (i.e. $7.92/2.464 = 3.2$) (lines 30-31).

Abstract: “We show that trends in the underlying ... will result ... healthcare intervention”. This is a modelling study and uncertainties included in future scenarios can be quite large. Thus, please remove the deterministic “will result” by “might result” given these uncertainties. You also need to mention the study region (... in Africa) as a naïve reader might think this is a global estimate.

We have clarified both these points (lines 30-33)

Intro – page 3: “global health threat and economic burden”

“to better forecast zoonotic disease risk and emergence”

“with a high case fatality rate” – true but we used to think that Ebola had a 80% mortality rate based on the old DRC outbreak data, while the observed mortality rate was far lower (about 50%) during the recent west African outbreak.

Fig 2 caption: The poverty-weighted case fatality rate for EBV is quite large (78%) while it was lower during the recent outbreak in West Africa (50ish %)

We agree this was confusing and have clarified the text to say this parameter is more appropriately considered as the maximum CFR. We agree CFR is hard to measure given detection biases and indeed a lower is figure for Ebola is likely to emerge with more data. Due to the poverty weighting in our model the mean CFR across the endemic area was 0.66, and therefore, closer to the 2014 than for published estimates

for the other ~20 outbreaks. We use country-specific data from Liberia, Guinea and Sierra Leone to set this relationship and therefore the values seen in those regions should be close to the values quoted by the reviewer. We have clarified all these points in the text (line 340).

Intro – page 5: “wild host ecology”

We are assuming that the reviewer wants use to change this and so have changed it to “host ecology” and hope that this addresses the concern raised (line 88).

Results – present day patterns: please be more descriptive about the African countries where EVD has not been reported yet (list them all e.g. Togo, Tchad, cote d’Ivoire, etc) as this is a valuable information.

Thank you for highlighting this important point – we predict that cases could reasonably spread to any country within west and central Africa. More importantly we show that Guinea-Bissau, Guinea, Liberia, Sierra Leone, Cote d’Ivoire, Ghana, Togo, Benin, Nigeria, Cameroon, Gabon, Equatorial Guinea, Congo, Democratic Republic of Congo, Central African Republic, Ruanda, Burundi, Uganda, Kenya and Tanzania all have the potential conditions to start outbreaks. Large outbreaks are most likely to start in Liberia, Sierra Leone, Southern Ghana and Nigeria, Eastern and Southern Democratic Republic of Congo, Uganda, Rwanda and Burundi. We have expanded the results to describe in more detail what the present day patterns show (lines 133-135).

Fig 3: Can you replace the black symbols (dots) which represent the locations of known EVD index cases to white dots or light grey crosses? Black on dark blue is difficult to read.

And: **Ref 2 Comment 13** –

10. In Figure 3, you cannot clearly see the black points. This should be changed in a revised version and statistical fitting should be done and included as part of this figure.

And: **Ref 3 Comment 9-**

Fig. 3 B - black symbols are difficult to see. Circles, triangles and tetrahedrons are not well resolved.

To improve the visibility of outbreaks and index cases we have substantially revised the figures and chosen a new colour scheme that also benefits from being colour-blind friendly.

Fig 4: Map of importation risk of EBOV infected individuals. Where is the keybar / scale ? The risk ranges from yellow to red but we have no idea whatsoever about the magnitude of that variable.

And: **Ref 2 Comment 14** –

11. In Figure 4, the resolution should be improved and a scale bar should be provided. For example, what is the difference in risk between red and orange?

We intended to show the pattern of relative risk in that figure but have now included a key to show the number of importations per run (Figure 4).

Page 11 – Future trajectories: “but increased by 14.7% under ...”. I calculated the surface ratios and it should be 13.9% (note that this might be related to rounding of the surface estimates). Please clarify.

This was not worded clearly – we have made this clearer and have checked the values quoted were correct (Line 238).

Fig 5: The authors should provide difference maps e.g. deltas calculated between the future risk maps and the baseline map shown on Fig 3 (these maps can be included in Sup. materials for example).

This is a good point - we have restructured figures to highlight this change (Figure 5).

Discussion: “ameliorate climate change”, prefer “mitigate climate change”.

We have removed this section in response to other reviewer comments.

Discussion: “it is currently unclear where vaccination should be targeted...”. Ok, but you plot risk maps so you could highlight countries at risk which were affected and countries where epidemics could occur (I guess this is the point of a risk model?)

We thank the reviewer for highlighting this point. We agree that our risk maps can help target preventative vaccination programmes and have amended the discussion to reflect this (lines 220-227)

Discussion: “vaccination estimated at \$135.90 per person”.

We have removed this section in response to other reviewer comments.

Discussion: The largest risk seems to be simulated over Nigeria. There is a paragraph in discussion mentioning this issue. The authors need to mention that Nigeria was slightly affected by Ebola; they had it under control rapidly, thanks to their public health services. This needs to be discussed in this paragraph: <https://europepmc.org/articles/pmc4584877> .

Also: **Ref 2 Comment 1** –

c. Their model predicts there is a strong spillover and subsequent outbreak risk in Nigeria and Ghana, two countries that the authors acknowledge have not reported Ebola outbreaks. The authors should provide justification for why their model produces these results. Given that they do not perform a proper validation exercise, this result makes the reader question the validity of their model and all subsequent results. Underreporting is not a strong enough justification for these results, particularly given that Nigeria and Ghana have two of the strongest healthcare systems in West Africa. These results imply that their model is not well-structured or parameterized to accurately capture Ebola dynamics.

We thank the reviewers for raising this important point and giving us a chance to discuss this more fully. The largest risk being in Nigeria makes sense in our ‘bottom up’ framework as there are a large numbers of bats, primates and people co-existing there. Given Nigeria is located in between Gabon/Congo and the site of 2014 outbreak in Guinea, there is currently a lack of evidence to discount Nigeria as being at some risk, especially given the current limited understanding of the disease. Unless there is empirical proof of an extreme discontinuity in the spatial distribution of seroprevalence, or of pathogenic strain distribution, we believe a sensible null expectation is that there are disease-carrying hosts present in Nigeria. While we isolate Guinea as a key source of large outbreaks, if this modelling exercise were done in 2013 it would seem this part of the output of the model was incorrect.

Furthermore, Ebola outbreaks are rare but potentially high-impact events and our knowledge of the ability for countries with functioning health care systems such as the

US, UK, Spain and Nigeria in controlling this disease consists of a handful of replications of a complex set of events. Regarding risks of secondary cases in Nigeria, we cannot simply assume the same outcome every time, especially given the thousands of different potential transport routes for infected people to enter Nigeria. We have updated the text to reflect these points (lines 231-253)

Methods: A few important things to clarify in this section:

1. The authors used the Gridded population of the world dataset v3 to obtain counts or densities of population. This dataset is available at 30arc sec (about 1km x 1km at the equator). Then they use an interaction sphere of 0.5m combined with daily walking distances to calculate geographic variation in human movement patterns. There is a spatial scale mismatch between this dataset and the approach (1km x 1km vs 0.5m – the 0.5 m radius for the whole of Africa might be super expensive computing wise as well). Please clarify, I might have been confused here

The 0.5 interaction sphere is used in a gas model to estimate the contact rate between people within each 1km grid square – we have clarified this point (line 425)

2. The climate model runs are only based on the UKMO model (Hadgem2-AO). The authors mention that this model is close to other climate models. However, changes in rainfall (and to a lesser extent temperature) can vary a lot across different climate models. As an example, for West Africa, it is difficult to draw any conclusions on the CMIP3 or CMIP5 climate model ensemble (some models predict drier conditions, other no changes, other simulate drier conditions). The authors need to mention that they did not include uncertainties related to the chosen climate model, and these can be quite large, in particular over the African continent.

We have extensively examined this issue and estimated the climatic niches of all the Ebola potential hosts for over 60 different climate models. We show the results in the Fig. 1 below. This analysis showed that outputs from Hadgem2-AO model result in a change in predicted host presence from 2010 to 2070 is central in relation to the overall variation seen from other models (Fig. 1). This model, therefore, represented a good working example for understanding climate impacts on EVD. We recommend further work examining how other models might impact future predictions (lines 293-296).

Figure 1. Plot of 68 climate models and the resultant predictions on Ebola host

presence. Each black dot shows the position of a different model in terms of a shift in the host niche's mean latitude (x-axis) or suitability change (y-axis). The red dot shows the position of the HadGem-AO model used in this analysis.

Reviewer #2 Comments

In “Modelling impacts of impacts of global change on emergence and epidemic potential of Ebola in Africa” the authors combine an epidemiological model with projected estimates of future climate, land use, and human population change to estimate the risk of both a spillover event from Ebola and the subsequent epidemic. Incorporating both of these factors in a single model is a fairly reasonable approach and it is reasonable to investigate how various socio-economic and climatic projects will impact a pathogen like Ebola.

We agree with the sentiment that considering both the environmental and human aspects of Ebola is reasonable approach to more fully capture its epidemiology.

Ref 2 Comment 1 –

1. The model should be validated against existing data.
 - a. It is unfair to use underreporting as justification to not perform a proper validation exercise. All diseases are underreported, however these data are still incredibly valuable and should be used to calibrate their model.
 - b. The only comparison of their model results with data are too casual and not backed up with a proper statistical analysis. This should be corrected in future versions of the manuscript.

AND: Ref 3 Comment 1-

On the other hand, lack of validation seems to be a major hurdle (as it is for many emerging disease problems) and there is very limited comparison with and referral to existing studies, particularly Ebola studies that make similar risk predictions, and the discussion is exceedingly short given the content of the ms. This makes it pretty difficult to know how well the model is really performing in its primary task, how much it departs from existing insights, and therefore how much of it will ultimately prove beneficial for EVD management over and above what is already available.

We have undertaken an extensive simulation exercise to understand whether observed case data is well represented by our model's risk predictions, and compared this to a randomised null result. We found that we our outputs have a much higher predictive ability to predict the case data (AUC = 0.83) compared to randomised versions of our risk layer (0.53 AUC, 10,000 randomisations, $p < 0.001$). This was true regardless of whether the Ebola 'endemic area' is set at $\pm 5^\circ$, $\pm 10^\circ$, $\pm 15^\circ$ or $\pm 20^\circ$, around the equator (figure S1).

- d. The result that the spillover risk is greater than the known area is unsurprising and the authors need to provide additional evidence that they are capturing known spillover areas and why their model is overestimating in these specific areas.

Given that many diseases in Africa suffer from chronic underreporting (Hotez et al. 2009) it would seem unreasonable to assume that Ebola does not suffer from the same biases. Our modelling results suggest that many single outbreaks occur and the chances of these being detected in remote areas is potentially very low as many serious febrile cases are wrongly attributed to Malaria (possibly up 2/3 of fevers; Dalrymple et al. 2017). We have discussed these points in more detail (lines 217-220).

References:

Hotez, P. J., Fenwick, A., Savioli, L. & Molyneux, D. H. Rescuing the bottom billion through control of neglected tropical diseases. *The Lancet* 373, 1570-1575 (2009).

Dalrymple, U., Cameron, E., Bhatt, S., Weiss, D.J., Gupta, S. and Gething, P.W., 2017. Quantifying the contribution of *Plasmodium falciparum* malaria to febrile illness amongst African children. *Elife*, 6, p.e29198.

e. The importation risk is highest in countries that have not experienced many importations of Ebola, even during the most recent large outbreak. Can the authors provide justification and further support for why their model is producing these unexpected and inconsistent results?

And: Ref 3 Comment 8-

P9 – “High risk of Ebola case importation using the current network of airline flights was seen in China, Russia, India, the **United** States as well as many high-income European countries (Fig. 4).” How do these results compare with observations of imported cases globally so far?

There have been only around 10 documented importations. As the flight network has thousands of nodes if we sampled it at random for just 10 cases the chances of recreating the observed pattern is highly unlikely. The pattern we report is effectively the shape of airline network when flying from common locations that experience outbreaks under our simulations (but sampled in much greater detail than the observed data). We have clarified this in the discussion and added in a comparison to the importations from the recent large outbreak (lines 145-149, 254-264). Future work could more accurately model this aspect if passenger number for each airline route were made available in the public domain.

Ref 2 Comment 2 –

2. The authors should assess the public health impact of these results.

a. In the introduction, the authors mention that this could be used to design adaptive vaccination programs, however they do not expand upon this in the main text. What are the implications of these results?

b. Also, vaccination in response to an Ebola outbreak will likely be reactive. Can the authors provide how their results will assist in this type of strategy?

c. It is not an actionable, helpful recommendation to reduce the population size. Could the authors provide more relevant recommendations?

We agree that have not discussed this in enough detail and have altered the discussion to focus on how our risk maps and model can be used in healthcare responses and how it can inform future vaccination plans. (lines 223-225)

Ref 2 Comment 3 –

3. The projection models require clarification.

a. Given the interdisciplinary nature of the journal, the authors need to provide the reader with additional information about the various project models used. Although these models may be very common in the climate change literature, they are likely unfamiliar to this audience. It would also be helpful if the authors created shorthand for each model (aside from the abridged name) to assist the reader in recalling the general assumptions/framework of the different model projections.

We appreciate this comment and clarification. As seen with many interdisciplinary analyses terminology often does not translate well. We have attempted to provide a

summary of models along with their formal definitions but have now given them easily remember informal names to aid reading (lines 109-118)

Ref 2 Comment 4 –

4. The results from the different projection models do not agree. To what extent can these results be used to draw general conclusions?

a. Often the results from the various climatic and socio-economic projection models produce contradictory results with a wide range of estimates (ex: decreases by 47% in some instances, but increases by 34% in other models). Given the inconsistent results, what is the main take away from these results?

b. In some instances, spillovers increase and in other instances they decrease. Given this, what can actually be interrupted from these results?

c. The projection models require additional clarification. For example, the authors use three different socio-economic development scenarios but do not provide enough detail about the differences between these scenarios.

We have reformatted Figure 5 to show the patterns of change for all cases across the range of scenarios tested. We have made it clear in results that changing the climate-change and SSP scenario have different relative impacts, which is a key message. From the new figure we can see a reduction in expected outbreaks (mainly due to SSP change and increasing health care effectiveness) in the “best case” scenarios but under “worst case” there is a widespread increase in cases (Figure 5). We have added in more detail about the SSP scenarios (lines 524-529).

Ref 2 Comment 5 –

1. Why do the authors project forward to 2070?

The reason we project forward is to understand the general direction of how future global change is likely to impact the underlying mechanisms of EVD infection. We project as far as the year 2070 to be able to detect in what direction climate change could have an effect, as we are only likely to have the power to detect any change by looking further into the future than the very near term. We have added in extra detail to clarify this (lines 282-288).

Ref 2 Comment 6 –

2. During the Ebola outbreak, health clinics that were specifically designed to treat the disease contributed to its control – these are inherently related to hospital infrastructure and health care treatment ability and should not be disregarded.

We agree that integrating hospitals remain a key future step for work of this type. At the present time is no formal analysis of location and areas serviced by every hospital and clinic in Africa, or a reasonable way of predicting how they would be set up in response to a major outbreak. It would certainly be interesting to examine this topic in more detail given realistic movement along the road networks to see how the speed of transmission would likely impact final epidemic size and duration of outbreak. We have added a comment on this to the discussion (lines 273-275, 329-334)

Ref 2 Comment 7 –

3. What is the motivation for the current grid scale? The authors perform a very detailed spatial analysis, but do not utilize these results. And if their overall results do not abide to current wisdom and are not calibrated to actual data, then what is the value of having such a fine resolution?

The reason the grid scale was chosen was due to the assumptions made about contact and movement. We wanted to use grid cell size small enough that a gas model of human movement could be credibly employed but still allowing fast computation (we have clarified this here – lines 336). The results presented are an emergent property of the interactions between hosts and people at the different spatial scales of the simulation.

Ref 2 Comment 8 –

4. How is healthcare incorporated into their model? Presumably it is incorporated in some way with the socio-economic projections, however it is currently unclear.

We have made the assumption that increasing or decreasing wealth under the different scenarios leads to a concomitant change in healthcare provisioning. We have clarified this (line 532).

Ref 2 Comment 9 –

5. Their model is fairly Ebola specific and it is unfair to state that it will be useful for other diseases.

We respectfully disagree with this statement. The modelling framework is disease agnostic with the host occurrence and disease transmission structure being completely flexible. For instance, the human-to-human aspect is only run, for instance, if human-to-human transmission is needed. All zoonotic diseases for which there are reasonable numbers of host occurrence points and some basic understanding of the starting parameters for the disease transmission parameters can be modelled. The main downside of our approach is currently the computational cost and we are actively working on this, to facilitate future analyses of other diseases. We have clarified this (lines 309-311).

Ref 2 Comment 10 –

6. They produced a stochastic model, but their results do not appear to include a lot of stochasticity. How was stochasticity incorporated?

The stochasticity was incorporated in the movement of infected people under different plausible starting conditions to examine the spread of the disease from different locations (clarified here lines 103-107)

Ref 2 Comment 11 –

7. On Page 4, the authors state that mechanistic models rarely incorporate spatial heterogeneous ecological and environmental information. This is untrue, particularly for vector-borne disease models.

Yes, we meant for zoonotic non-vector borne diseases – yes there is much work in the literature, especially for mosquitos, and we have clarified the meaning of the statement (line 67-69).

Ref 2 Comment 11 –

8. In Figure 3, what is the unit of the values? For example, the values go up to 0.01, but is this a percentage?

This is the proportion of times there was a case in a cell per run – we have clarified this here (line 743-746)

Ref 2 Comment 12 –

9. The authors consider an epidemic if there is at least 1500+ cases, what is their epidemiological (not purely statistical) justification? How do their results change when this

arbitrary value is varied?

This is an interesting question. While it changes the intensity of the result it does not change the pattern or reported regions of concern, so we decided to report patterns arising from a single threshold.

Epidemic threshold = 1500

Epidemic threshold = 200,000

Ref 2 Comment 15 –

12. In Figure 5, if the proportion of time is equal to 0 (shown in dark blue), why is it included in the area projections? And corresponding figure?

We thanks the reviewer for spotting this error. Grey is equal to zero and this should be indicated on the key – we have rectified this on the new figures.

Ref 2 Comment 16 –

13. In Figure 6, can the authors provide an interpretation of these values? Is it greater than expected, as in 20x or a 20% increase? What is the significance value? What is the statistical test?

It is a chi-square test and the standardised residuals are the difference in the observed count (i.e. runs that assigned to each category), minus the expected count, divided by the standard deviation of the expected count. As this is fairly standard statistical term we have not added in much more detail but have clarified that it is the relative contribution of the different categories to chi statistic and added in more detail about the test into the legend.

Ref 2 Comment 17 –

14. What is the relationship between wealth and R_e over time?

The R_e value is parameterised by wealth and wealth changes over time according to the SSP scenario. We have added in detail to clarify this (lines 532).

Ref 2 Comment 18 –

15. What is the airline data used in the analysis? Could the actual destination data and weights based on the frequency of flights and/or passenger sizes be used?

And: Ref 2 Comment 19 –

16. This analysis assumes that the population will increase, but the road and airline networks

will not. Is this reasonable?

And:Ref 3 Comment 16-

P23 - "...with the probability of travel by air per day ϵf , being proportional to the total number of flights per day divided by the population of that country". Could this assumption overestimate importations, since economic restrictions would make it difficult to afford flights to USA/Europe etc.? Maybe an overestimate to assume that the whole population is 'available' to travel?

Only people close to airports (roughly within one day's reasonable journey) can travel on planes and, therefore, infected people have to move on road networks to airports before they can fly. It is the probability of travel per person that uses a country-level estimate, so those people in less travelling countries have a lower chance of flying when near airports compared to those near airports in high income countries. Given that everyone in the model is assigned the mean wealth of the country we cannot easily make differentiations on what flights people can and can't afford.

We argue that the current shape of airline network, however, is an emergent property of the wealth, trade and tourism relationships between countries. Poorer countries with low tourist flows will have fewer flights to richer countries. Conversely, poor countries with high tourism will need to have more flights to richer countries to transport tourists. The model we run does not speak to nationality of the person flying - but an infected person acting as a disease carrier on an aircraft might be a tourist returning to a home nation or a country-national flying. We could weight the probability of travelling on a flight by distance travelled but it is likely that this would be an oversimplification of reality (i.e. there would be likely higher numbers than expected of flights between South America and the Iberian peninsula) and would need a more complex model based on more data than we have available presently. We have added in some text to describe this process (lines 478-483).

Regarding future transport network expansions: Currently we employ a simple model where every road and every flight has the same capacity and probability. More complexity may be needed to accurately capture future movement patterns, but the lack of an obvious path to make future predictions of these patterns, mean that we chose not to implement this step in the current study. For instance, it could be that roads nearer the capital are preferentially widened or that all roads within each country are brought up to certain capacity before others are widened. Similarly, with airlines, most major cities are already connected and predicting how the capacity may be altered to reflect demand, or changing world trade patterns, is difficult. We have added in more detail to reflect this (lines 533-536).

Ref 2 Comment 20 –

17. Why is the model run for a single year?

We calculated the number of outbreaks expected per year and calibrated the model so that approximately the same rate of successful contacts per year occur. We recognise this is essentially an arbitrary number but wanted to allow sufficient time for outbreaks to initiate and run the course through to the point that there were no more cases.

Reviewer #3 Comments

The contribution from Redding et al., is undoubtedly an impressive and multifaceted piece of

work. It is very ambitious in its aims of integrating mechanistic with correlative modeling approaches for the analysis of Ebola virus disease (EVD) outbreaks. The paper contributes predictions of present day risk for EVD outbreaks and is then combined with scenario-based forecasting to evaluate the way that environmental and social changes expected to 2070 may influence this risk. This could be very useful for imagining management responses and decision making on preventive measures. Importantly, the contribution attempts to close a number of significant gaps in our knowledge regarding the connectivity and inter-dependence of a large range of socio-ecological processes that together contribute to EVD's risk profile.

We thank the reviewer for their positive comments and agree on the importance of understanding socio-economics in disease modelling.

Ref 3 Comment 2-

P4 – “One downside is that mechanistic models rarely incorporate spatially heterogeneous ecological and environmental information, such as the known high variance of bat abundance and pathogen sero-prevalence across widespread individuals.” – it's not clear how/whether the correlative components used address these two example issues.

We thank the reviewer for highlighting this. We have changed the examples to something that our modelling framework does address (Line 69).

Ref 3 Comment 3-

P4 – “but no studies that we are aware of have considered whole- system analyses for major epidemic zoonoses, such as Ebola.” – what about other diseases? Not entirely clear from this statement how novel systems analyses are for infectious diseases more broadly.

As far as we are aware systems models are quite rare for zoonotic diseases. We have added this context into the discussion (lines 73-76).

Ref 3 Comment 4-

P4 – “Like other rare or poorly- sampled diseases, Ebola suffers from limited data availability, meaning pattern-finding, correlative analytical techniques are at a disadvantage” – for demonstration of this novel systems approach, especially extending to scenario-based projections, is EVD really the best test case for this reason? Although I completely agree with the premise, there seem to be two somewhat competing agendas here: demonstration of the novel integrative method, and improving models for EVD risk assessment. The former may be better served by looking at a disease with greater availability of data for adequate validation. The latter arguably requires the former to be maximally convincing for data poor applications. The projections make a third key deliverable of the ms, but this is arguably undermined until both the first and second points are addressed. It's exciting, but just a massive amount of uncertainty to squeeze into a single paper.

We argue that we are examining the outputs of a fairly simple ‘null’ model here: Essentially, we define the potential ‘arena of contact’ of people and hosts (and then stochastically examine the fates of infected people). Much of the movement and contact parameters are relative here (i.e. we are constraining them to actual case numbers rather than trying to recreate them) and the results we choose to report i.e. the spatial patterns and shape of disease distributions are will be relatively insensitive to much of the further modelling complexity. Having said that, we have successfully predicted all the subsequent outbreaks since the modelling was undertaken and the ongoing one in Kivu is in an area that is highlighted by us as at risk of a large outbreak (1500+ cases).

We argue that Ebola is good candidate for our ‘null’ modelling approaches as it is clear

that waiting for more Ebola cases, to be able to use more traditional epidemiological methods, would be highly costly to those societies involved. The future will tell us how accurate our approach is, but our results can provide one possible answer to highlight research areas to proactively investigate in the future.

Lastly, arguably the most important result here, one that can direct much future work, is the need to find out more about the spatial discontinuity of seroprevalence and/or strain severity to establish whether large, well connected populations in Ghana and Nigeria are indeed at high risk or not (Lines 237-243).

Ref 3 Comment 5-

P4 - The authors say that there is a “trade-off between increasing human populations and loss of reservoir host species through anthropogenic land-use conversion”. However, the presence of fruit-bat species suspected to be linked to the Ebola virus, such as *Eidolon helvum* (Haiman et al. 2010, PLOS ONE 5: e11978, though not considered by the authors) seem to have responded positively to land-use conversion from forest to agriculture, and even keep large urban colonies (Peel et al. 2017, Acta Chiropterologica 19: 77-92). In addition, two recent papers in Scientific Reports document links between recent deforestation and increases in EVD risk (Rulli et al., 2017; Olivero et al., 2017).

We did not include *Eidolon helvum* when we considered which bat species to include, based on the weight of evidence at the time. Specifically, the studies we cite show that there was very low seroprevalence in sampled *E. helvum* and from this information we concluded that *E. helvum* was either a rare or accidental host.

We also note that the quoted statement by the reviewer is misleading due to being oversimplified. Using our approach both negative and positive land use change effects are experienced and, as a result, any species that benefit from human-dominated land use would indeed do so in our analysis. We have corrected the statement accordingly (line 94).

Ref 3 Comment 6-

P8 - “the at-risk area for EBOV-EVD is much larger than the areas known to have experienced disease outbreaks thus far”— at the risk of appearing self-promoting, a number of other studies have come to the same conclusion and could be referred to here and results compared/contrasted (e.g.,: Pigott et al. 2016, eLIFE 5:e16412; Olivero et al. 2017, Mammal Review 47:24-37; Murray et al. 2015, PNAS 112:12746-12751;). That countries or parts of countries that could be at risk that have never experienced Ebola outbreaks and therefore may be insufficiently prepared is an important message coming through from all of the above studies, including this one.

We have added in more background to explain these differences and any support of previous analyses (lines 133-139; 209-210).

Ref 3 Comment 7-

P8 – “Our risk map also identified areas that are endemic for the other EVD strains, likely due to similar transmission pathways and reservoir host characteristics (Fig. 3A).” Does this indicate a lack of specificity for the target organism due to e.g., lack of specific knowledge of hosts/reservoirs or some other factors among strains? If so, perhaps there is an argument for including information on the other strains too in the model building and validation processes?

While this was not the intention, it appears that ecological transmission routes have a similar spatial pattern across the Ebola strains. Including point locations from other

strains in our validation does result higher accuracy scores. Having not targeted these strains we do feel that we can comment any further than we have already, except noting the similarities and ability of our outputs to capture those case locations.

Ref 3 Comment 10-

Fig 4 – what defines the EVD endemic zone? North Africa but not e.g., South Africa is indicated, while Fig 3 shows most of North Africa as being unsuitable while at least a bit of South Africa being suitable?

This is a really interesting question as it is not clear what defines the endemic zone and this an area that our study can very much examine. In the validation exercise we use three potential limits $\pm 5^\circ$, $\pm 10^\circ$, $\pm 15^\circ$ around the equator as illustrated in figure S1. We have added in more detail (lines 128-129).

Ref 3 Comment 11-

Fig S3 – unclear how to interpret the violin plot, it seems like a qualitative validation step? Can more formal validation tests be conducted to help with interpretation regarding model performance benchmarked against observations?

Thanks for highlighting this point. The violin plots serve two main purposes – to show that the observed and simulated results have similar distributions and also to show that a more complete sampled distributions potentially has an as yet unsampled area of very large outbreaks. We also have now added in a validation analysis (please see comments above).

Ref 3 Comment 12-

P16 - Nigeria is identified here as a key area (in fact the most important area) for epidemics to be initiated and with potential for many small outbreaks. This is in contrast to previous studies (e.g., Peterson et al. 2004, Emerging Infectious Diseases 10: 40-47; Pigott et al. 2016, eLIFE 5:e16412; and Olivero et al. (2017, Mammal Review 47:24-37). Again, it would be beneficial to discuss the results in light of predictions from previous studies.

The aim of the present study was to create a model that does not use case data and instead explains Ebola risk using a bottom-up approach. Several of the studies highlighted by the reviewer use case data to pattern-match the environmental conditions that are similar to areas where outbreaks have occurred. Such an approach is, by its nature, not likely to identify areas that are very different to areas with known outbreaks. However, the site of the 2014-2016 index case was in area remote from the previous set of outbreaks, so there is a risk correlative analyses not challenged with data point will report this area as not high risk. We have further discussed this in the discussion (lines 231-242).

Ref 3 Comment 13-

P19 - The outputs from models for different species (first reservoirs, then non-reservoirs, and finally reservoirs with non-reservoirs) are combined. These outputs seem to be probability values. The presence probability of a species is influenced by environmental conditions, but also by the prevalence of its distribution within the study area. Consequently, probability values for species with different prevalence are not comparable. It is thus important to include here an explanation of how these models were combined.

Each species we consider important in the transmission cycle has an independent layer where probability of occurrence is estimated for each grid cell. We assume that perfect mixing occurs within each grid cell and people meet a species with the probability of it being there. Therefore, one minus the probability of the host being there, under these

assumptions, multiplied across all species that have a non-zero prediction for that grid cell, is the probability of not meeting any species. One minus this value gives the probability of meeting at least one species in each cell. We have clarified this further (lines 385-390)

Ref 3 Comment 14-

P19 - When combining models for different species, the authors “down-weight the probability of non-reservoir occurrence by two thirds and reservoir occurrence by one third”. This decision is based on the fact that “only a third of index cases were attributed to non-reservoir host spill-overs”. However, none of the index cases recorded up to now has been proved to be a result of contacts with species presumed to be the reservoir (i.e. bats). This justification may thus require reformulation.

We presume, from weight of literature, that bats are the ‘working’ reservoir host and, as a regular bushmeat target and competitor for fruit crops, that contact rate is relatively high. We considered apes/duikers as secondary routes for transmission so did not want to count their presence equally. In reality, especially for apes, their suitable areas are quite narrow compared to bats and restricted to already bat-suitable forests and this weighting will only have minimal impact, so we did not choose to sensitivity test it. We have clarified this in the methods (lines 390-396).

Ref 3 Comment 15-

P20 - "We assumed that β for the Infectious and Funeral compartments was equivalent, due to the contact rates of moving individuals in the Infectious compartment being offset by large aggregations of individuals at funerals." This seems like quite an assumption. It was not clear if this was addressed through sensitivity analyses.

We agree that this represents a sizeable assumption but believe that in our context it is reasonable. This is due to the contact rate model used here being very simple (i.e. gas movement) and in order to differentiate pre-death and post-death behaviour we would need to substantially change the movement model to something more akin to an agent-based model, where individuals can have unique patterns of movement. If we were to use a more complex model (i.e. not a gas model) then we agree it would be unreasonable to conflate these two compartments, but this would provide too higher computational cost for this modelling exercise. This is clarified in the methods (Lines 410-414).

Ref 3 Comment 17-

P23 – The method of constructing projected land-use cover and change seems pragmatic. However, this may be perceived as a major oversimplification for those in the trade of land-use change modeling, and both its accuracy and integration with other elements of the projections do not appear to have been tested/validated. As the authors acknowledge, at coarser resolutions, there are existing products available that provide harmonized land use change projections. It’s not clear what the objection to using the existing harmonized dataset really is from the description, it could be the resolution on number of land cover classes or the spatial resolution. In either case, more description is required, and it would be informative to do some basic benchmark testing of the bespoke analysis against these or other similar products (e.g., via downscaling) in order to get some basic information about its robustness. Furthermore, a robust (or even indicative) 5x5km land-use change product for the region and period indicated would be a novel contribution in itself and of wide potential application and interest. However, it might also be worth considering the value of sticking to the 5x5km resolution in the first place if this can’t reasonably be represented for key datasets in the analysis.

Ideally, we would have used a land-use product at the appropriate resolution but we emphasise the main issue here is not with spatial resolution but that of resolution of the habitat classes themselves. The land-use harmonisation datasets (LUH) assign land cover classes in terms of human land-use rather than natural habitat. Thus, a square designated as primary in the LUH dataset might be an arctic habitat or a tropical rainforest - offering completely different conditions for species to survive in. This makes this dataset not applicable to understanding how a species' probability of occurrence changes. This point, and more details of the process, has been added into methods (lines 502-506).

Regarding validation – this is not a straight-forward process. There are no future habitat-level prediction datasets out there (that we know of – but they are in the pipeline) to validate against and the LUH dataset just has land-use, not land-cover. To determine if the two datasets correspond to each other one would need to work out what each of the land-use LUH grid cell designations (e.g. primary, secondary) mean in terms of what habitat was there in the first place, and for this one would need to look at to, for instance, historical MODIS data, so this quickly becomes circular. Unless there is a clear way forward here, projecting forward the MODIS trends and checking that current patterns of deforestation and agricultural expansion continue as expected, seems the most pragmatic way forward.

References

1. Li, S.-L. *et al.* Essential information: Uncertainty and optimal control of Ebola outbreaks. *Proceedings of the National Academy of Sciences* **114**, 5659-5664 (2017).
2. Alexander, K. A. *et al.* What factors might have led to the emergence of Ebola in West Africa? *PLoS neglected tropical diseases* **9**, e0003652 (2015).
3. Pigott, D. M. *et al.* Mapping the zoonotic niche of Ebola virus disease in Africa. *Elife* **3**, e04395 (2014).
4. Lau, M. S. *et al.* Spatial and temporal dynamics of superspreading events in the 2014–2015 West Africa Ebola epidemic. *Proceedings of the National Academy of Sciences* **114**, 2337-2342 (2017).
5. Kramer AM, *et al.* (2016) Spatial spread of the West Africa Ebola epidemic. *Royal Society Open Science* 3(8):160294.
6. CIESIN (2013) Global Roads Open Access Data Set, Version 1 (gROADSv1). (NASA Socioeconomic Data and Applications Center (SEDAC), Palisades, NY).
7. World Bank (2014) *World Development Indicators* (World Bank, Washington DC, USA).

Reviewers' Comments:

Reviewer #1:

Remarks to the Author:

Review of "Modelling impacts of global change on emergence and epidemic potential of Ebola in Africa"

Manuscript number: NATCOM-18-16450A

Authors: Redding et al., 2018

Recommendation: Minor revisions

General:

The paper greatly improved since the last round of submission. I just have a few final minor comments.

Minor points:

POINT 1:

In response letter and the text - the authors mention 67 different climate models (L294-296)

This is related to different climate model versions overall (5 different versions of the UKMO climate model for example)

The overall ensemble of GCMs is usually around 30ish

<https://portal.enes.org/data/enes-model-data/cmip5/resolution>

Please provide full list of climate models somewhere in Supp Material

POINT 2: Brief statement about where EBV outbreaks could spread in the abstract would be valuable (no need to describe all countries but you can say that most countries in West Africa and Central Africa could be affected)

Generic comment: I spotted a few minor grammatical and spelling mistakes to double check (see later comments)

L105: "which areas of the world are most at risk"

Ok but the global risk assessment mainly relies on flight data

Most of the detailed risk modeling work is carried out for Africa so perhaps:

"unsampled effects of Ebola epidemiology in Africa, and potential spread of EBOV at global scale based on flight data."

L142-L146: This is a long and confusing sentence - please reword

L182-L186: This sentence is very confusing. Please reword

L226-228: Also as the ... - something is missing in that sentence or remove "as the"

L232: "We identify Nigeria (but also Ghana, Kenya and Rwanda) for many small outbreaks to occur..."

L254: "would be incorrect"

L262: "in terms of..."

L307: "Efforts to decrease poverty..." or "Decreases in poverty...", please double check grammar and

typos

Figure 1 - could you specify which risk factors are only varying in space and other risk factors varying both in space and time?

Other Minor comments:

Figures in the pdf exceed the A4 format - make sure this is fine with the editorial office during the proofing stages.

Remove track changes in the Supp Material document please.

Reviewer #3:

Remarks to the Author:

I am satisfied with the authors efforts to revise the ms according to the comments provided.

Response to reviewers' comments: NCOMMS-18-16450A

We thank the reviewers for this final round of comments and provide our responses (in red) to suggestions below:

Reviewer #1 (Remarks to the Author):

POINT 1:

In response letter and the text - the authors mention 67 different climate models (L294-296)
This is related to different climate model versions overall (5 different versions of the UKMO climate model for example)

The overall ensemble of GCMs is usually around 30ish

<https://portal.enes.org/data/enes-model-data/cmip5/resolution>

Please provide full list of climate models somewhere in Supp Material

Thanks for pointing this out. The value 67 was typo from double counting models for apes, bats and duikers. When we corrected this the number of unique models was 32. We have added in a list in Supplementary Table 5 (SI).

POINT 2: Brief statement about where EBV outbreaks could spread in the abstract would be valuable (no need to describe all countries but you can say that most countries in West Africa and Central Africa could be affected)

Change made as requested (line 26).

Generic comment: I spotted a few minor grammatical and spelling mistakes to double check (see later comments)

L105: "which areas of the world are most at risk"

Ok but the global risk assessment mainly relies on flight data

Most of the detailed risk modeling work is carried out for Africa so perhaps:

"unsampled effects of Ebola epidemiology in Africa, and potential spread of EBOV at global scale based on flight data."

Change made as requested (line 100-101).

L142-L146: This is a long and confusing sentence - please reword

Change made as requested (line 142-146).

L182-L186: This sentence is very confusing. Please reword

Altered to make clearer (line 182-186).

L226-228: Also as the ... - something is missing in that sentence or remove "as the"

Altered to make clearer (line 222).

L232: "We identify Nigeria (but also Ghana, Kenya and Rwanda) for many small outbreaks to occur..."

Change made as requested (line 227).

L254: "would be incorrect"

Typo corrected (line 257)

L262: "in terms of..."

Typo corrected (line 291)

L307: "Efforts to decrease poverty..." or "Decreases in poverty...", please double check grammar and typos

Typo corrected and manuscript re-read through (line 302)

Figure 1 - could you specify which risk factors are only varying in space and other risk factors varying both in space and time?

Change made as requested (line 713-718).

Reviewer #3 (Remarks to the Author):

I am satisfied with the authors efforts to revise the ms according to the comments provided.

We thank the reviewer for their previous comments.

Having made these changes we hope that the manuscript is now ready for final acceptance.

Kind regards,

David Redding (for all authors)